# Accelerometer-derived physical activity and mortality in individuals with type 2 diabetes

Zhi Cao[1,2,6], Jiahao Min[1,6], Han Chen[1], Yabing Hou[3], Hongxi Yang[4], Keyi Si [5,7] & Chenjie Xu [1,7]

Physical activity (PA) has been shown to reduce diabetes mortality, but largely based on imprecise self-reported data, which may hinder the development of related recommendations. Here, we perform a prospective cohort study of 4003 individuals with type 2 diabetes (T2D) from the UK Biobank with a median follow-up of 6.9 years. Duration and intensity of PA are measured by wrist-worn accelerometers over a 7-day period. We observe L-shaped associations of longer duration of PA, regardless of PA intensity, with risks of all-cause and cancer mortality, as well as a negatively linear association with cardiovascular disease mortality. 18.8%, 28.0%, and 31.1% of deaths are attributable to the lowest level of light-intensity PA, moderate-intensity PA, and vigorous-intensity PA, respectively. Collectively, our findings provide insights for clinical guidelines that should highlight the potential value of adherence to greater intensity and duration of PA for patients with T2D.

Diabetes affects around 529 million adults worldwide, directly resulting in 1.7 million deaths in 2021[1]. The mortality risk in diabetic adults was ~60% higher than that in the non-diabetic[2]. Physical activity (PA) has been reported as an indispensable factor in lowering or even canceling out this excess mortality risk. However, more physical inactivity was observed in diabetic adults [Diabetes (U.S. Centers for Diseases Control and Prevention); https://www.cdc.gov/diabetes/index.html][3], partly because they may face more physiological and psychological barriers to PA[4,5]. Their complicated conditions may also bias the benefit from PA[6]. Investigating the association between PA and mortality specifically in diabetic adults is vital to inform PA recommendations tailored to this high-risk population. A meta-analysis of 12 cohorts has indicated that the all-cause mortality risk in the highest PA category was 40% lower than that in the lowest among diabetic adults, but the exact volume of PA cannot be specified because its grouping rules are highly heterogeneous across cohorts[7]. Moreover, some prospective cohort studies reported that ≥150 min/week of moderate-to-vigorous-intensity PA (MVPA) or ≥75 min/week of vigorous-intensity PA (VPA) was associated with 14%–37% lower risks of all-cause mortality in diabetic adults[8–10]. However, all abovementioned studies are based on self-reported PA, which is prone to recall bias and insensitiveness in capturing light-intensity PA (LPA) and is usually restricted to limited PA domains[11]. In contrast, accelerometer is a promising wearable device that can objectively record an individual's intensity and duration of PA in free-living conditions for a period of time[12,13]. It may help uncover the true magnitude of the association between PA and mortality and facilitate personalized T2D management[13]. A harmonized meta-analysis of eight studies found that in the general population the maximal risk reduction in all-cause mortality for accelerometer-measured MVPA (about 60%) was about twice the magnitude as reported in studies relying on self-reported PA[14]. In addition, existing studies have demonstrated that objectively measured PA could be a potential predicator for cardiovascular disease (CVD) mortality and cancer mortality in the general population[15,16]. Nevertheless, accelerometry-based evidence in diabetic adults was scant. Two studies derived from the NHANES, a study from the Look AHEAD Trial and a study from the Walking Away from Type 2 Diabetes trial all demonstrated significant risk reductions in related to all-cause mortality with higher volumes of PA[17–20]. However, the sample sizes of these accelerometry-based research among type 2 diabetes (T2D) adults were <2000; accelerometers were worn on the hip or waist, which may omit

[1]School of Public Health, Hangzhou Normal University, Hangzhou, China. [2]School of Public Health, Zhejiang University School of Medicine, Hangzhou, China. [3]Yanjing Medical College, Capital Medical University, Beijing, China. [4]School of Basic Medical Sciences, Tianjin Medical University, Tianjin, China. [5]School of Public Health, Shanghai Jiao Tong University School of Medicine, Shanghai, China. [6]These authors contributed equally: Zhi Cao, Jiahao Min. [7]These authors jointly supervised this work: Keyi Si, Chenjie Xu. e-mail: sikeyi0219@163.com; xuchenjie@hznu.edu.cn

activities that primarily involve upper body movement; and dose–response analyses were not performed. To fill the knowledge gaps, we aimed to investigate the dose–response associations between the duration of PA in different intensities and risks of all-cause, cancer, and CVD mortality among adults with T2D (96% of all diabetes cases[1]) in the UK Biobank, the largest prospective cohort with PA measured by wrist-worn accelerometers to date (Fig. 1). Furthermore, we investigated the potential effect modification by factors related to diabetes severity, such as glycemic control, diabetes duration, and diabetes medication use.

## Results

### Population characteristics

As shown in Supplementary Fig. 1, of the 502,401 UK Biobank participants, 32,709 had T2D at baseline, of whom 4604 had accelerometry data. After excluding those who had insufficient wear time (n = 312),

daylight saving time shifts during wear period (n = 174), and missing information on covariates (n = 115), 4003 participants with T2D were left in the main analysis, with a mean age of 64.9 years (standard deviation, 6.9 years) and 63.0% were males. Diabetic participants included in the analyses were generally healthier than those who were excluded, with a higher level of education, a smaller body mass index (BMI) and waist circumference, a lower prevalence of smoking, and better health status at baseline (Supplementary Table 1). Similarly positive trends were seen in the included diabetic participants with higher volumes of PA (Table 1 and Supplementary Tables 2–4).

### Association of physical activity with all-cause mortality

During a median follow-up of 6.9 years (interquartile range 6.3–7.4 years), 339 death cases were documented. Regardless of PA intensity, the dose–response associations of PA duration with all-cause mortality

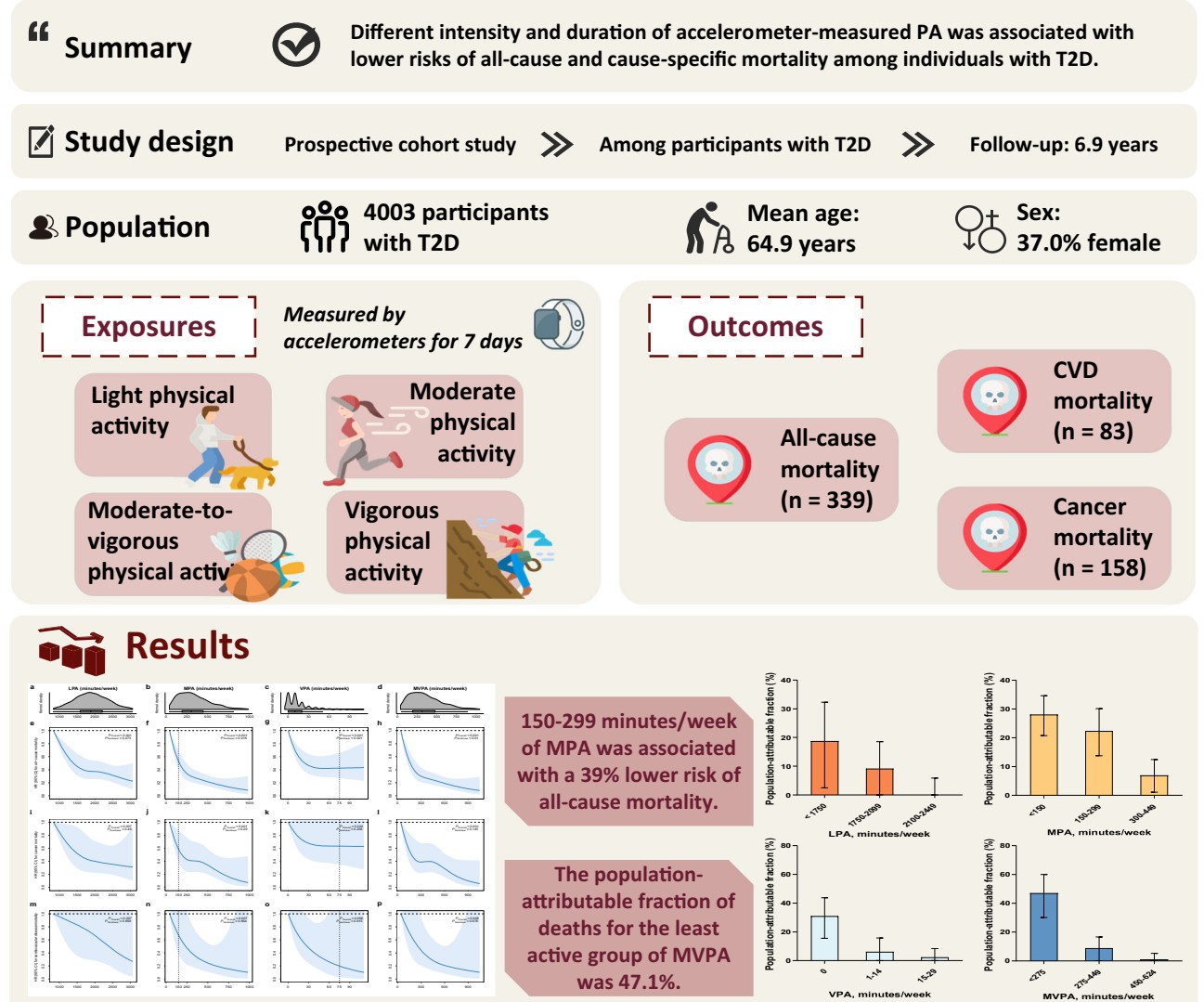

**Fig. 1 | Summary of the study.** The study includes 4003 participants with type 2 diabetes from the UK Biobank to explore the association between accelerometer-derived physical activity and all-cause and cause-specific mortality. The top three boxes present the main findings, the study design, and the baseline characteristics of the population of this study. Below them, we assessed the exposure and the outcome variables in this study. The physical activity data in this study were obtained from accelerometer measurements and classified into four intensities. The outcomes of interest were all-cause mortality, cancer-cause mortality, and cardiovascular-cause mortality. The number of cases for each outcome is provided. Bottom left, restricted cubic splines fitted in the Cox models were used to examine the dose–response associations of physical activity with all-cause and cause-specific mortality. Bold lines represent HRs, while shaded areas indicate 95% CI. Bottom right, we estimated the population-attributable fractions for all-cause mortality. Bottom middle presents the main results of the study. The widths of the lines extending from the center points represent 95% CI. HR hazard ratio; CI confidence interval; PA physical activity; LPA light-intensity physical activity; MPA moderate-intensity physical activity; VPA vigorous-intensity physical activity; MVPA moderate-to-vigorous-intensity physical activity; T2D type 2 diabetes; CVD cardiovascular disease; PAFs population-attributable fractions.

**Table 1 | Baseline characteristics of 4003 participants by MVPA**

| Characteristics | Total | Device-measured MVPA, min/week | | | |
| --- | --- | --- | --- | --- | --- |
| | | <275 | 275-449 | 450-624 | ≥625 |
| Total, n | 4003 | 1665 | 1197 | 684 | 457 |
| Age, year, mean (SD) | 64.9 (6.9) | 66.8 (6.1) | 64.5 (6.9) | 63.3 (7.2) | 61.2 (7.3) |
| Sex, male, n (%) | 2523 (63.0) | 1044 (62.7) | 769 (64.2) | 427 (62.4) | 283 (61.9) |
| Ethnicity, White, n (%) | 3740 (93.4) | 1587 (95.3) | 1129 (94.3) | 609 (89.0) | 415 (90.8) |
| Education, college or university, n (%) | 1332 (33.3) | 492 (29.5) | 418 (34.9) | 255 (37.3) | 167 (36.5) |
| BMI, kg/m², mean (SD) | 31.1 (5.8) | 32.5 (6.1) | 31.0 (5.7) | 29.7 (4.8) | 28.1 (4.8) |
| Waist circumference, cm, mean (SD) | 102.0 (14.4) | 105.8 (14.3) | 101.6 (13.7) | 98.7 (13.0) | 93.7 (13.6) |
| Smoking status, n (%) | | | | | |
| Never | 1760 (44.0) | 650 (39.0) | 560 (46.8) | 327 (47.8) | 223 (48.8) |
| Former | 1905 (47.6) | 842 (50.6) | 541 (45.2) | 311 (45.5) | 211 (46.2) |
| Current | 338 (8.4) | 173 (10.4) | 96 (8.0) | 46 (6.7) | 23 (5.0) |
| Diet score, mean (SD) | 3.7 (1.9) | 3.6 (2.0) | 3.7 (1.8) | 3.8 (1.9) | 4.1 (2.0) |
| Sleep score, mean (SD) | 2.8 (1.0) | 2.7 (1.0) | 2.9 (1.0) | 2.9 (1.0) | 3.0 (1.1) |
| Alcohol intake, g/day, mean (SD) | 16.8 (21.6) | 14.9 (20.8) | 17.5 (22.3) | 18.3 (21.8) | 20.0 (21.7) |
| LPA, min/week, mean (SD) | 1901.3 (478.0) | 1648.1 (437.4) | 1990.2 (408.8) | 2125.5 (418.5) | 2255.7 (387.6) |
| MPA, min/week, mean (SD) | 340.8 (206.5) | 163.8 (64.1) | 340.3 (47.5) | 498.2 (55.4) | 751.2 (169.0) |
| VPA, min/week, mean (SD) | 16.6 (27.5) | 4.3 (7.9) | 15.0 (15.1) | 27.3 (28.7) | 49.7 (52.3) |
| Season of wear, n (%) | | | | | |
| Spring | 848 (21.2) | 336 (20.2) | 247 (20.6) | 164 (24.0) | 101 (22.1) |
| Summer | 1108 (27.7) | 434 (26.1) | 338 (28.2) | 205 (30.0) | 131 (28.7) |
| Autumn | 1152 (28.8) | 491 (29.5) | 355 (29.7) | 168 (24.6) | 138 (30.2) |
| Winter | 895 (22.4) | 404 (24.3) | 257 (21.5) | 147 (21.5) | 87 (19.0) |
| Wear duration, day, mean (SD) | 6.7 (0.7) | 6.7 (0.7) | 6.7 (0.6) | 6.7 (0.7) | 6.6 (0.7) |
| Diabetes duration, year, mean (SD) | 11.2 (9.4) | 11.5 (9.1) | 10.8 (8.8) | 11.0 (10.3) | 11.8 (10.6) |
| HbA1c, mean (SD) | 50.9 (13.0) | 51.7 (13.2) | 50.4 (13.2) | 49.9 (11.7) | 50.8 (13.6) |
| Insulin medication use, n (%) | 601 (15.1) | 235 (14.1) | 171 (14.3) | 98 (14.4) | 97 (21.3) |
| Self-rated health, n (%) | | | | | |
| Excellent | 200 (5.0) | 61 (3.7) | 52 (4.3) | 52 (7.6) | 35 (7.7) |
| Good | 1851 (46.2) | 634 (38.1) | 599 (50.0) | 357 (52.2) | 261 (57.1) |
| Fair | 1476 (36.9) | 688 (41.3) | 421 (35.2) | 233 (34.1) | 134 (29.3) |
| Poor | 476 (11.9) | 282 (16.9) | 125 (10.4) | 42 (6.1) | 27 (5.9) |
| History of cancer or CVD, n (%) | 1201 (30.0) | 628 (37.7) | 338 (28.2) | 168 (24.6) | 67 (14.7) |
| History of hypertension, n (%) | 2308 (57.7) | 1128 (67.7) | 647 (54.1) | 321 (46.9) | 212 (46.4) |
| Long-standing illness, disability or infirmity, n (%) | 2980 (74.4) | 1335 (80.2) | 864 (72.2) | 482 (70.5) | 299 (65.4) |
| Illness, injury, bereavement, or stress in last 2 years, n (%) | 1962 (49.0) | 851 (51.1) | 585 (48.9) | 326 (47.7) | 200 (43.8) |

*BMI* body mass index, *CVD* cardiovascular disease, *LPA* light-intensity physical activity, *MPA* moderate-intensity physical activity, *VPA* vigorous-intensity physical activity, *MVPA* moderate-to-vigorous-intensity physical activity, *SD* standard deviation.

were L-shaped (Fig. 2). At the inflection points, associations were observed with all-cause mortality reductions of ~60% at 1800 min/week for LPA, 70% at 300 min/week for moderate-intensity PA (MPA), 45% at 30 min/week for VPA, and 70% at 300 min/week for MVPA, compared with the least active one percentile. Additional associations of all-cause mortality risk reductions followed by a longer duration of PA were limited for VPA (additional 1% by as long as 100 min/week) but significant for LPA (15% by 3000 min/week), MPA (20% by 1000 min/week), and MVPA (20% by 1000 min/week). Accordingly, when the duration of PA was classified into four levels, the risk for all-cause mortality gradually decreased as the duration increased to a higher level, irrespective of PA intensity (p value for trend <0.001, Table 2 and Fig. 3). For example, compared with <150 min/week of MPA, the multivariable-adjusted hazard ratios (HRs) and 95% confidence intervals (CIs) for 150–299, 300–449, and ≥450 min/week were 0.61 (0.47-0.79), 0.41 (0.29-0.56), and 0.24 (0.15-0.36), respectively.

The shorter the current PA duration, the higher the proportions of death cases that could be prevented if replaced with the highest level of PA duration (Fig. 4 and Supplementary Table 5). For example, the population-attributable fractions (PAFs) for mortality sequentially decreased with higher durations of MVPA, with PAFs of 47.1%, 8.8%, and 1.0% for those engaging <275 min/week, 275–449 min/week, and 450–624 min/week, respectively. Similar patterns to MVPA were observed for other PA intensities, with 18.8%, 28.0%, and 31.1% of PAFs for the lowest activity groups of LPA, MPA, and VPA, respectively.

**Association of physical activity with cause-specific mortality**

A total of 158 cancer deaths (46.6% of all deaths) and 83 CVD deaths (24.5% of all deaths) were recorded. Irrespective of PA intensity, L-shaped associations with PA duration were found for cancer mortality whereas negative linear associations were observed for CVD mortality (Fig. 2). Generally, more risk reductions were associated with higher levels of PA (Fig. 3, Supplementary Tables 6, 7). Compared with

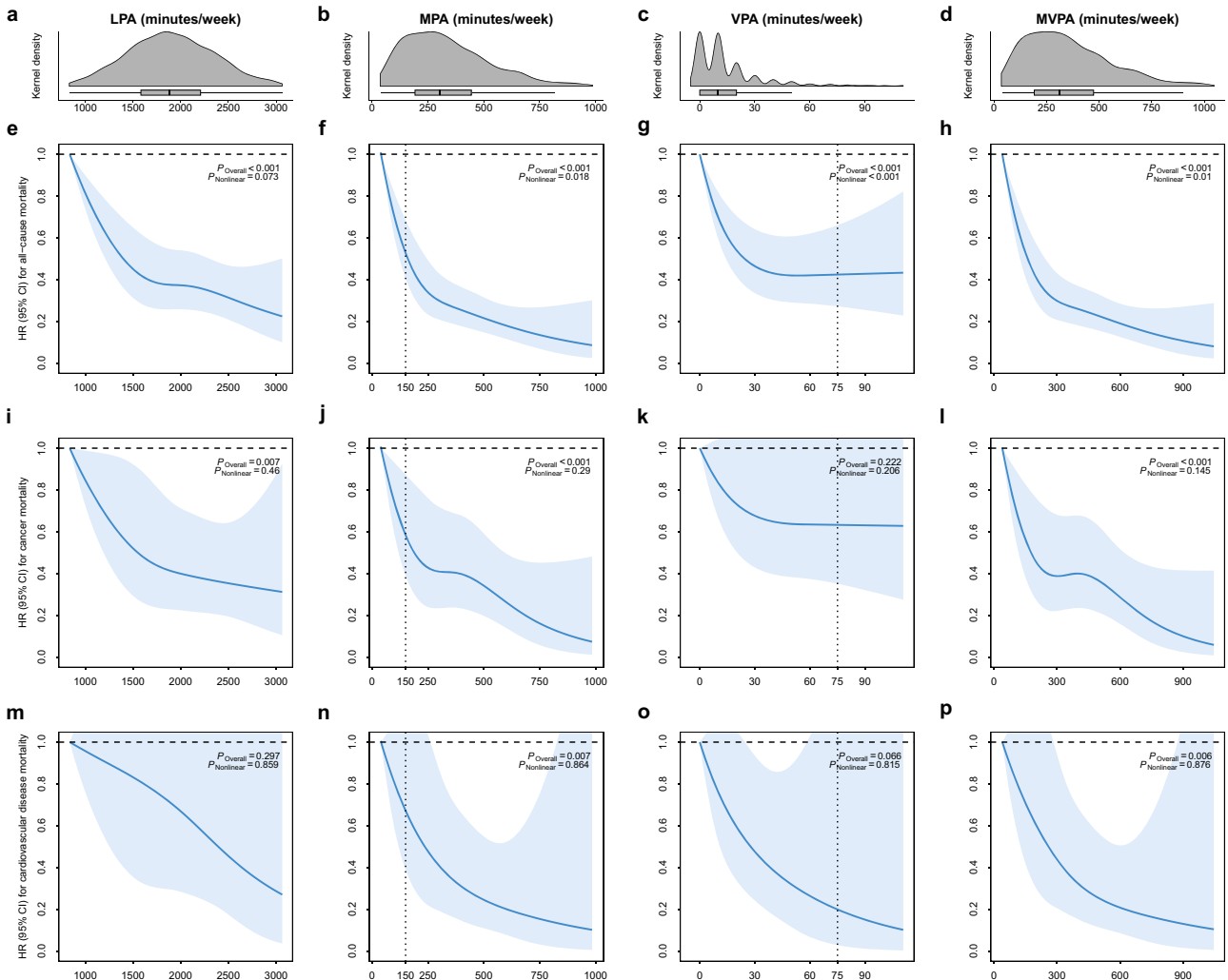

**Fig. 2 | Dose–response association between accelerometer-derived physical activity and all-cause and cause-specific mortality among participants with type 2 diabetes. a–d** The gray density histograms depict the distribution of LPA, MPA, VPA, and MVPA, respectively. The data depicted in the box and whisker plots span from the minimum to the maximum values (min/week). The lower and upper boundaries of the box correspond to the 25th and 75th percentiles, respectively, and the central vertical lines within the boxes represent the median values. **e–h** Dose–response association of LPA, MPA, VPA, and MVPA with all-cause mortality. Bold lines represent HRs, while shaded areas indicate 95% CI. **i–l** Dose–response association of LPA, MPA, VPA, and MVPA with cancer mortality. Bold lines represent HRs, while shaded areas indicate 95% CI. **m–p** Dose–response association of LPA, MPA, VPA, and MVPA with CVD mortality. Bold lines represent HRs, while shaded areas indicate 95% CI. All adjusted for age (years), sex (male or female), ethnicity (white or others), education (college/university or others), season at the time of accelerometry recording (spring, summer, autumn, or winter), accelerometer wear duration (days), smoking status (never, former, or current), alcohol intake (g/day), diet score (0–7), sleep score (0–5), body mass index (kg/m²), waist circumference (cm), self-rated health (excellent, good, fair, or poor), long-standing illness, disability or infirmity (yes or no), illness, injury, bereavement, or stress in last 2 years (yes or no), history of cancer or cardiovascular disease (yes or no), history of hypertension (yes or no), and diabetes duration (years). Wald tests were used in the analyses to obtain the two-sided *p* values. HR hazard ratio, CI confidence interval, LPA light-intensity physical activity, MPA moderate-intensity physical activity, VPA vigorous-intensity physical activity, MVPA moderate-to-vigorous-intensity physical activity. Source data are provided as a Source Data file.

<150 min/week of MPA, ≥450 min/week of MPA was associated with a 63% (95% CI: 33%–80%) and 68% (95% CI: 21%–87%) lower risk of cancer and CVD mortality, respectively. Meanwhile, the HRs with VPA ≥ 30 min/week were 0.61 (95% CI: 0.32–1.17) for cancer mortality, and 0.35 (95% CI: 0.10–1.15) for CVD-cause mortality, compared with none of VPA.

### Joint association between physical activity and mortality
The risk matrix illustrating the joint associations between combinations of PA with different intensities and mortality was shown in Fig. 5 based on HRs from Supplementary Tables 8–10. We found that almost all combinations of PA were associated with lower risks of all-cause and cause-specific mortality. Different combinations of PA could associate with similar risk reductions in all-cause mortality, such as

300–449 min/week of MPA and 2100–2449 min/week of LPA (HR = 0.35; 95% CI: 0.19–0.63), 1–14 min/week of VPA and 2100–2449 min/week of LPA (HR = 0.33; 95% CI: 0.17–0.63), 1–14 min/week of VPA and 300–449 min/week of MPA (HR = 0.35; 95% CI: 0.22–0.54).

### Stratified analyses and sensitivity analyses
No statistically significant interactions were found between all PA intensities and age, sex, BMI, waist circumference, smoking status, alcohol intake, diet scores, sleep scores, and history of hypertension (*p* value for interaction >0.05, Supplementary Tables 11–14). The results were largely consistent with the main analyses when participants with poor self-rated health status were excluded (Supplementary Table 15), when further adjusted for or stratified by diabetes severity factors (Supplementary Tables 16,

**Table 2 | Hazard ratios (95% CI) for all-cause mortality according to physical activity among individuals with type 2 diabetes**

| Exposures | No. of cases | Incidence rate per 1000 person-year | Hazard ratio (95% CI)[a] | | |
|---|---|---|---|---|---|
| | | | Model 1 | Model 2 | Model 3 |
| LPA (min/week) | | | | | |
| <1750 | 167 | 16.66 | 1.00 (Ref) | 1.00 (Ref) | 1.00 (Ref) |
| 1750-2099 | 102 | 12.93 | 0.82 (0.64,1.05) | 0.83 (0.65,1.06) | 0.90 (0.70,1.16) |
| 2100-2449 | 45 | 8.03 | 0.53 (0.38,0.74) | 0.55 (0.40,0.77) | 0.60 (0.43,0.84) |
| ≥2450 | 25 | 7.48 | 0.53 (0.35,0.82) | 0.55 (0.36,0.84) | 0.60 (0.39,0.93) |
| *p-value* for trend | | | <0.001 | <0.001 | =0.001 |
| MPA (min/week) | | | | | |
| <150 | 119 | 28.46 | 1.00 (Ref) | 1.00 (Ref) | 1.00 (Ref) |
| 150-299 | 129 | 15.03 | 0.58 (0.45,0.75) | 0.59 (0.46,0.76) | 0.61 (0.47,0.79) |
| 300-449 | 61 | 8.60 | 0.36 (0.26,0.49) | 0.38 (0.27,0.52) | 0.41 (0.29,0.56) |
| ≥450 | 30 | 4.28 | 0.20 (0.13,0.31) | 0.21 (0.14,0.32) | 0.24 (0.15,0.36) |
| *p-value* for trend | | | <0.001 | <0.001 | <0.001 |
| VPA (min/week) | | | | | |
| 0 | 183 | 20.91 | 1.00 (Ref) | 1.00 (Ref) | 1.00 (Ref) |
| 1-14 | 89 | 9.97 | 0.53 (0.41,0.68) | 0.54 (0.42,0.70) | 0.58 (0.44,0.75) |
| 15-29 | 48 | 8.30 | 0.46 (0.33,0.64) | 0.48 (0.34,0.66) | 0.52 (0.37,0.73) |
| ≥30 | 19 | 5.57 | 0.35 (0.22,0.57) | 0.37 (0.23,0.61) | 0.43 (0.26,0.70) |
| *p-value* for trend | | | <0.001 | <0.001 | <0.001 |
| MVPA (min/week) | | | | | |
| <275 | 228 | 21.00 | 1.00 (Ref) | 1.00 (Ref) | 1.00 (Ref) |
| 275-449 | 73 | 9.01 | 0.48 (0.37,0.63) | 0.51 (0.39,0.66) | 0.54 (0.41,0.71) |
| 450-624 | 26 | 5.50 | 0.32 (0.21,0.48) | 0.33 (0.22,0.49) | 0.36 (0.24,0.55) |
| ≥625 | 12 | 3.78 | 0.25 (0.14,0.45) | 0.27 (0.15,0.49) | 0.30 (0.16,0.54) |
| *p-value* for trend | | | <0.001 | <0.001 | <0.001 |

*HR* hazard ratio, *CI* confidence interval, *LPA* light-intensity physical activity, *MPA* moderate-intensity physical activity, *VPA* vigorous-intensity physical activity, *MVPA* moderate-to-vigorous-intensity physical activity.

[a]Hazard ratios (95% CI) were calculated in Cox proportional hazards model: model 1, adjusted for age (years), sex (male or female), ethnicity (white or others), education (college/university or others), season at the time of accelerometry recording (spring, summer, autumn, or winter), and accelerometer wear duration (days); model 2, further adjusted for smoking status (never, former, or current), alcohol intake (g/day), diet score (0 to 7), and sleep score (0 to 5) based on model 1; model 3, further adjusted for body mass index (kg/m$^2$), waist circumference (cm), self-rated health (excellent, good, fair, or poor), long-standing illness, disability or infirmity (yes or no), illness, injury, bereavement, or stress in last 2 years (yes or no), history of cancer or cardiovascular disease (yes or no), history of hypertension (yes or no), and diabetes duration (years) based on model 2. Wald tests were used to obtain the two-sided *p-value*.

17), when we additionally adjusted for T2D-related complications (Supplementary Table 18), when LPA, MPA, and VPA were mutually adjusted in models (Supplementary Table 19 and Supplementary Fig. 2), when missing data were imputed by chained equations (Supplementary Table 20), when the competing risk regression model was used (Supplementary Table 21), when the volume of PA was expressed in metabolic of equivalent task (MET) (Supplementary Table 22 and Supplementary Fig. 3), and when participants were grouped as those who meet the current guideline PA recommendations and those who did not (Supplementary Table 23). The associations were slightly attenuated in some categories when we excluded participants who had cancer or CVD at baseline (Supplementary Table 24), when we excluded participants who died during the first 2 and 4 years of follow-up (Supplementary Table 25), or when cancer deaths were stratified into T2D/obesity-related cancers or -independent cancers (Supplementary Table 26).

## Discussion

In this prospective cohort study of 4003 adults with T2D, we found that irrespective of PA intensity, longer duration of accelerometer-measured PA was significantly associated with a lower risk of all-cause, cancer, and CVD mortality, without minimal or maximal threshold. Up to 56.9% deaths among T2D patients could potentially be associated with performing <625 min/week of MVPA. The associations were independent of many potential confounders, including those related to diabetes severity.

Prior accelerometry-based studies specific to T2D patients were scant but all demonstrated significant risk reductions in mortality-related outcomes with higher volumes of PA, such as a 29% lower risk of premature all-cause mortality for every 60-min increase in daily ambulatory movement in the NHANES and a 2.6% lower risk of a composite CVD outcome including all-cause mortality for every 100 MET·min/week increase in MVPA from baseline to 4 years in the Look AHEAD Trial[17–19]. However, the magnitude of these associations cannot be directly compared with ours because the outcomes were slightly different and the NHANES did not account for PA intensity. In contrast, there have been many relevant studies using self-reported PA. Our accelerometry-based results reinforced the non-linear inverse association between PA and mortality risk in T2D patients observed in these studies, but with larger effect sizes. In our study, a 70% of risk reduction in all-cause mortality was seen at the inflection point of MVPA (300 min/week), larger than the 55% at 500 min/week by self-report in a US study of 41,726 diabetic adults[21]. Moreover, we found that every 600 MET·min/week increase of accelerometer-measured PA was associated with 11% lower risk of all-cause mortality in patients with T2D, much larger than 4% as reported by a meta-analysis of six prospective cohort studies (N = 32,221)[22]. Similarly, 50% risk reduction of all-cause mortality was associated with ≥150 min/week of MPA in our study, much larger than 14% from another study using self-reported PA in the UK Biobank[9]. The between-study difference might be attributed to the different background of the study populations and the different measurement methods of PA. Accelerometry represents an improvement over self-report by reducing recall bias, misclassification of

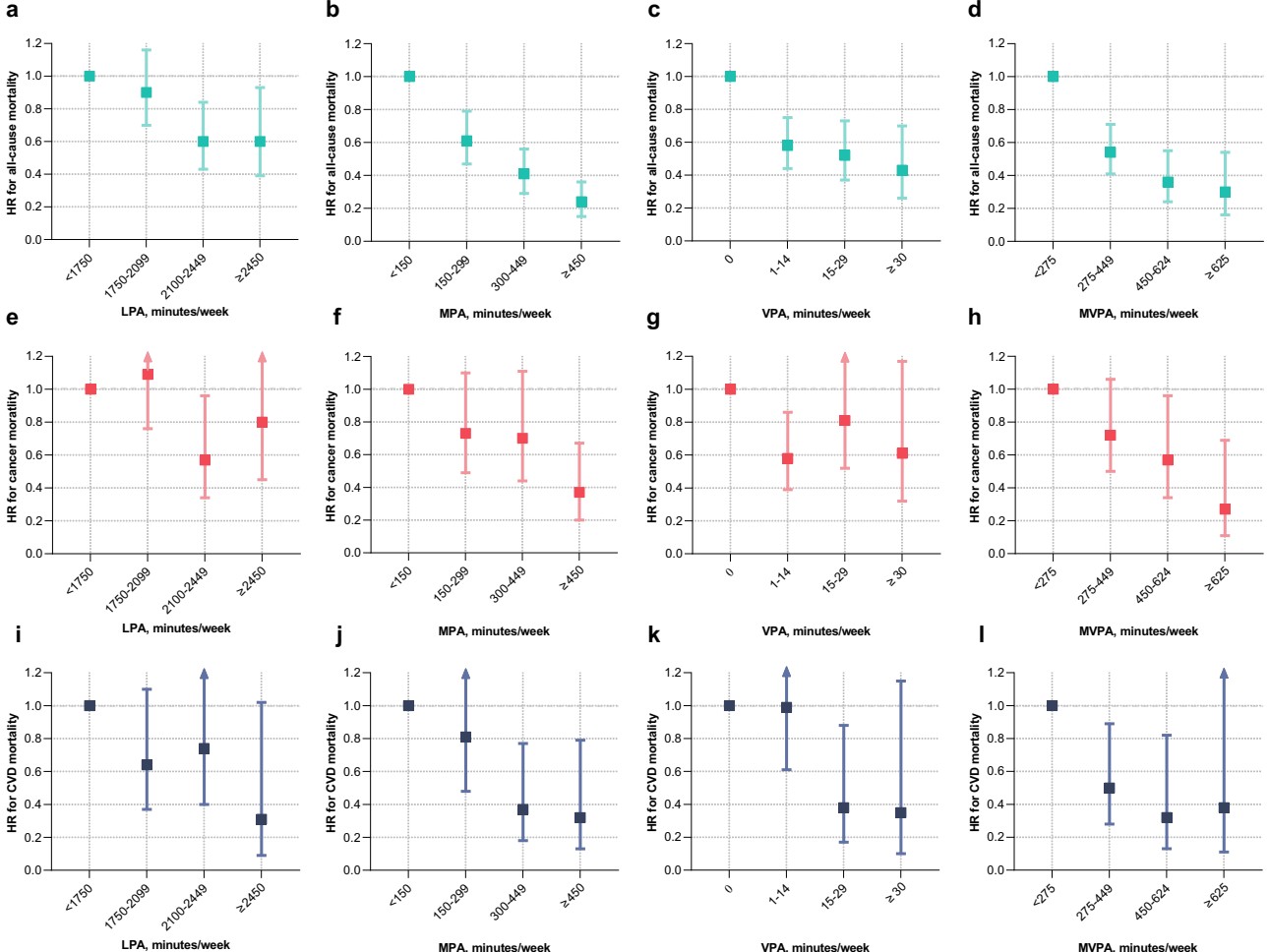

**Fig. 3 | Association between accelerometer-derived physical activity and all-cause and cause-specific mortality among participants with type 2 diabetes.** We utilized diabetic participants (n = 4003) from the UK Biobank with valid accelerometer data in the analyses. **a**–**d** Associations of LPA, MPA, VPA, and MVPA with all-cause mortality. **e**–**h** Associations of LPA, MPA, VPA, and MVPA with cancer mortality. **i**–**l** Associations of LPA, MPA, VPA, and MVPA with CVD mortality. All analyses were adjusted for age (years), sex (male or female), ethnicity (white or others), education (college/university or others), season at the time of accelerometry recording (spring, summer, autumn, or winter), accelerometer wear duration (days), smoking status (never, former, or current), alcohol intake (g/day), diet score (0–7), sleep score (0–5), body mass index (kg/m²), waist circumference (cm), self-rated health (excellent, good, fair, or poor), long-standing illness, disability or infirmity (yes or no), illness, injury, bereavement, or stress in last 2 years (yes or no), history of cancer or cardiovascular disease (yes or no), history of hypertension (yes or no), and diabetes duration (years). Data are presented as HR ± 95% CI for Cox regression. Error bars represent the 95% CIs for each effect estimate. Arrows indicate that the CI are outside the range shown in the figure. Source data are provided as a Source Data file. HR hazard ratio, CI confidence interval, LPA light-intensity physical activity, MPA moderate-intensity physical activity, VPA vigorous-intensity physical activity, MVPA moderate-to-vigorous-intensity physical activity.

intensity, omission of fragmented PA, and insensitiveness in capturing LPA.

The L-shaped dose–response associations of accelerometer-measured LPA and MVPA with all-cause mortality in T2D patients in our study mirrored those in the general population in a harmonized meta-analysis, with similar inflection points for LPA (around 2000 min/week) but different ones for MVPA (T2D: 300 min/week; general: 140 min/week)[14]. The magnitude of risk reduction was smaller in T2D patients than in the general from 1400 min/week to 2100 min/week of LPA (26% vs. 50%) and from around 0 to 140 min/week of MVPA (55% vs. 60%), but beyond this level, the risk continued to decrease in T2D patients while levelled off or even slightly increased in the general. Such discrepancy suggests a longer duration of PA to achieve the same amount of risk reduction in mortality but a larger maximal risk reduction in T2D patients and indicates the necessity of PA recommendations tailored for T2D patients.

The latest guideline recommends ≥150 min/week of MPA to most T2D patients or ≥75 min/week of VPA to patients who are younger and more physically fit[23]. These recommendations were primarily made based on evidence from self-reported PA, which may be optimized by incorporating evidence from accelerometer-measured PA. In our study, the risk for all-cause mortality decreased sharply as the duration of MPA increased to 300 min/week with a less drastic magnitude thereafter. A dramatic risk reduction was also seen in VPA till 30 min/week and then became steady. This finding suggests that the effort required to increase the volume of MPA may not be proportional to the benefit after reaching 300 min/week, while the recommended threshold of VPA can be lowered to 30 min/week, which can bring considerable benefit and is easier to implement.

Despite the fact that regular PA is a recognized modifiable protective factor for T2D prognosis, accelerometry data in 871 T2D patients showed that 55.8%, 57.4%, and 34.9% of White, African American, and Hispanic individuals did not meet the recommended threshold in guidelines[24]. In line with the statement of the World Health Organization that "every step count", we found that any amount of PA,

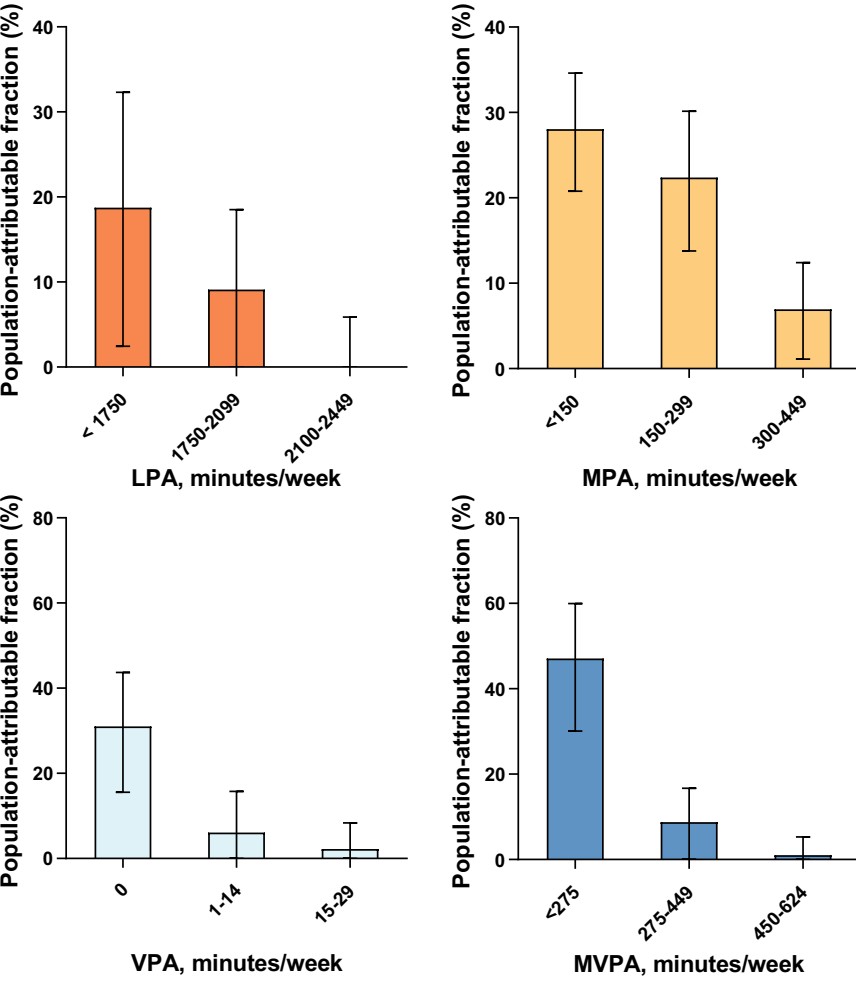

**Fig. 4 | Population-attributable fractions for all-cause mortality associated with accelerometer-derived physical activity among participants with type 2 diabetes.** We utilized diabetic participants (n = 4003) from the UK Biobank with valid accelerometer data in the analyses. Population-attributable fractions below 0 were truncated at 0. LPA light-intensity physical activity, MPA moderate-intensity physical activity, VPA vigorous-intensity physical activity, MVPA moderate-to-vigorous-intensity physical activity. The widths of the lines extending from the center points represent 95% CI. Source data are provided as a Source Data file.

even in light intensity, was associated with lower mortality risks versus none among T2D patients. For the least active individuals or those who cannot bear MVPA, starting with small volumes of low-intensity PA may be more feasible and easier to stick with. Another noteworthy point is that as short as 15 min/week of VPA have been linked to a 40% lower risk of mortality that needs to be achieved by 125 min/week of MPA. This short-duration-large-benefit activity is worth trying but some T2D patients may need to consult with health professionals before performing VPA since VPA may be difficult for some people to adopt and maintain or even cause discomfort for older, less fit patients with T2D[25]. Consultation with health professionals prior to performing VPA may be appropriate. For T2D patients who have just reached the threshold of the guideline, more PA is encouraged to obtain more survival benefit, as long as their functional ability allows. The risk matrix in Fig. 5 can be used as a reference when T2D patients want to choose a combination of PA with different intensities based on their own ability and interest.

The underlying biological mechanisms for the association between PA and mortality reduction in T2D patients can be summarized as PA-induced improvement on insulin sensitivity, glycose control, lipoprotein profile, blood pressure, and blood coagulability, which are likely to reduce the risk of diabetic complications and CVD[26]. In addition, a marked increase in cytokines, such as interleukin-6 and

interleukin-10, provoked by PA can protect against chronic inflammation in T2D[27].

The major strengths of this study were its large-scale sample size and rich data sources that enabled us to accurately identify patients with T2D. Furthermore, PA was objectively measured by an accelerometer which can capture leisure and non-leisure PA across multiple domains[28] and minimized the recall bias and reporting bias. However, there are several limitations that still need to be considered. First, there is currently a lack of robust evidence on whether a 7-day measurement is representative of habitual PA. A previous validation study showed a 7-day measurement was strongly associated with PA over a period of up to 3.7 years[29]. If the measurement error is random, the true association between PA and mortality may be underestimated in our study. Second, we utilized wrist-worn accelerometers that are more accurate in discriminating different intensities compared to waist-worn accelerometers[30]. However, they may not fully capture activities involving minimal arm movement, such as cycling, that ankle-worn accelerometers can quantify more accurately. Third, we only focused on the duration and intensity of PA, but the other properties of PA, such as fragmentation metrics, diurnal pattern, and distributional pattern may be equally important, which warrant further investigation. Fourth, the cut-points for various PA intensities in this study are consistent with

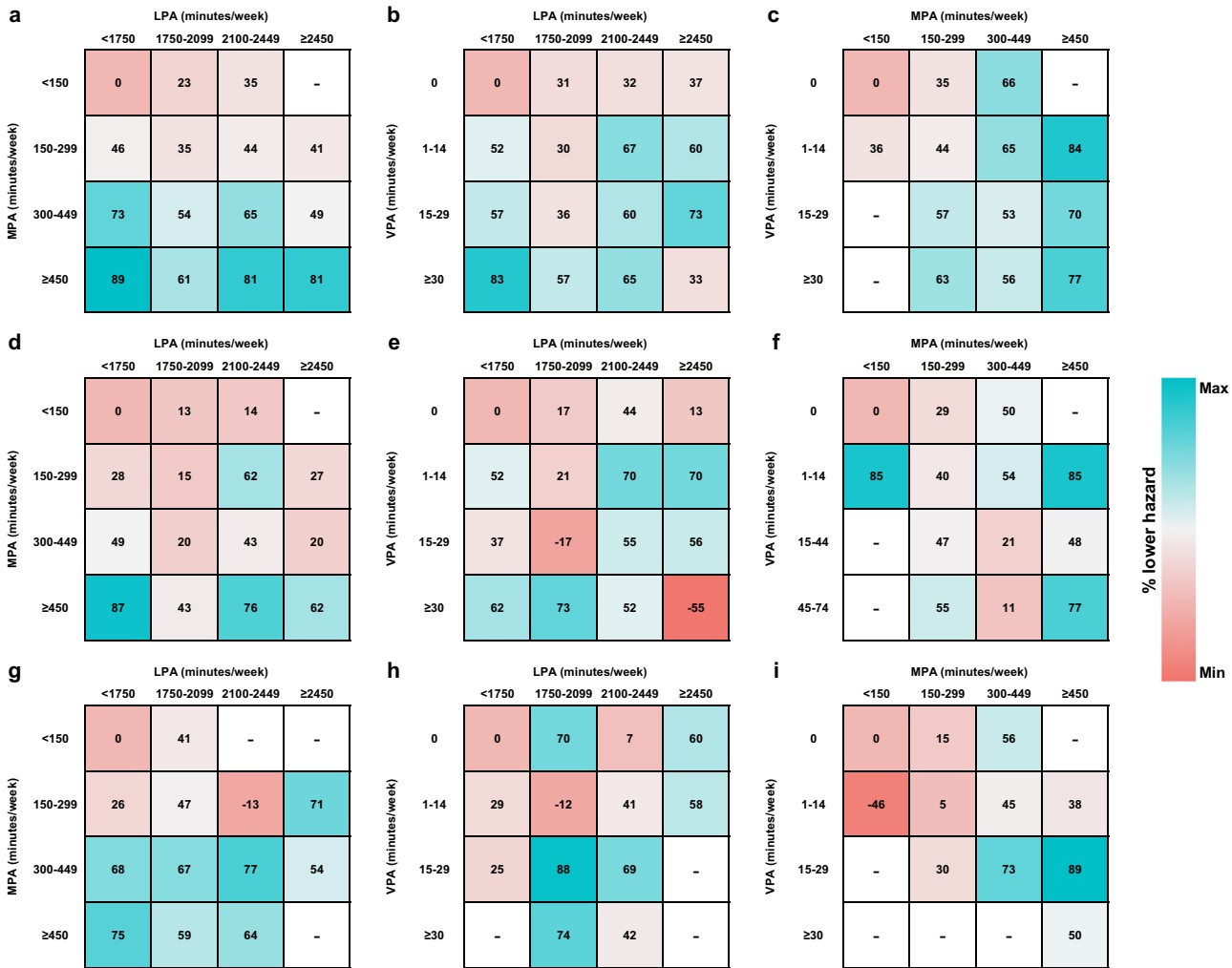

**Fig. 5 | The risk matrix for the joint association of accelerometer-derived physical activity with all-cause and cause-specific mortality among participants with type 2 diabetes. a** The joint associations of LPA and MPA with all-cause mortality. **b** The joint associations of LPA and VPA with all-cause mortality. **c** The joint associations of MPA and VPA with all-cause mortality. **d** The joint associations of LPA and MPA with cancer mortality. **e** The joint associations of LPA and VPA with cancer mortality. **f** The joint associations of MPA and VPA with cancer mortality. **g** The joint associations of LPA and MPA with CVD mortality. **h** The joint associations of LPA and VPA with CVD mortality. **i** The joint associations of MPA and VPA with CVD mortality. All analyses were adjusted for age (years), sex (male or female), ethnicity (white or others), education (college/university or others), season at the time of accelerometry recording (spring, summer, autumn, or winter),

accelerometer wear duration (days), smoking status (never, former, or current), alcohol intake (g/day), diet score (0–7), sleep score (0–5), body mass index (kg/m²), waist circumference (cm), self-rated health (excellent, good, fair, or poor), long-standing illness, disability or infirmity (yes or no), illness, injury, bereavement, or stress in last 2 years (yes or no), history of cancer or cardiovascular disease (yes or no), history of hypertension (yes or no), and diabetes duration (years). The numbers presented are the associated reduction in hazard (percentage) compared to the least active group, with a redder color indicating a lower risk reduction and a bluer color representing a higher risk reduction. There was not a sufficient number of participants to estimate hazard ratios in the blanked cells. Source data are provided as a Source Data file. LPA light-intensity physical activity, MPA moderate-intensity physical activity, and VPA vigorous-intensity physical activity.

most studies involving accelerometry data from the UK Biobank. However, it is important to note that these thresholds, which can be subject to variability due to device and demographic differences, may introduce a degree of subjectivity. Future research should, therefore, utilize more objective methods, such as data-driven approaches based on machine learning, to differentiate between PA intensities. Fifth, as for any observational studies, we cannot exclude the role of residual/unmeasured confounding or make causal conclusions, even though we have adjusted for a wide range of potential confounders. Sixth, PA could be influenced by participants' health status. Although the results were similar to the main analyses after excluding those with poor self-rated health status and those who died within the first 2 or 4 years of follow-up, the possibility of reverse causation cannot be fully eliminated. Seventh, though the sample size of the UK Biobank was as large as 0.5 million, the response rate was only 5.5% and accelerometer-measured PA was

only available for a subset of participants. Diabetic adults in these analyses were generally healthier, more physically active, and had a lower mortality rate than the overall diabetic adults. The selection bias may limit the generalizability of the current findings. Future studies are recommended to verify these associations between PA and mortality in T2D patients across various national populations, with longer follow-up time and larger sample size of diabetic adults. Nonetheless, recent evidence suggests that any potential bias resulting from poor representativeness would have a minimal impact on the association of PA with mortality in the UK Biobank[31].

In conclusion, inverse associations previously observed between the duration of self-reported PA and the risk of mortality in individuals with T2D were verified in the UK Biobank using accelerometry data, but with a larger magnitude, which informs an update of the recommended level of MPA and VPA, and an addition of recommendations on LPA in future guidelines.

## Methods

### Study design and population

The datasets used in this study were obtained from the UK Biobank (Application: 79095), which received approval from the North West Multi-centre Research Ethics Committee (R21/NW/0157), and the Biomedical Research Ethics Committee of Hangzhou Normal University (200400001). All participants gave written informed consent and have signed the informed consent to be linked to national electronic health-related datasets[32]. The UK Biobank is a large, population-based prospective cohort, in which more than 0.5 million participants aged 37–73 years attended 1 of the 22 assessment centers across England, Scotland, and Wales to complete touchscreen questionnaires, physical examinations, and biological sample collections between 2006 and 2010 (baseline of the UK Biobank). Then between 2013 and 2015 (baseline of this study), 240,000 invitations were randomly sent to them for PA measurement by accelerometers, with a response rate of 44%. Devices were dispatched for 106,053 participants and, of these, data were received from 103,666. In this study, participants with T2D at baseline were identified through an integration of multiple data sources: self-reported T2D, insulin use or glycated hemoglobin (HbA$_{1c}$) level ≥48 mmol/mol according to the American Diabetes Association criteria, and code E11 in the electronic health records (England and Wales: Health Episode Statistics; Scotland: Scottish Morbidity Records) according to the 10th Revision of the International Classification of Diseases (ICD-10). The algorithms for the capture and adjudication of prevalent T2D was consistent with previous research[9].

### Device-measured physical activity

Participants were asked to wear Axivity AX3 (Newcastle upon Tyne, UK) triaxial accelerometers on their dominant hands for 7 days, when the sensor captured the acceleration at 100 Hz with a dynamic range of ±8 g (unit of gravity)[13]. Participants with insufficient wear time (<72 h), poor device calibration, or daylight saving time shift during their wear period were excluded in this study. Minutes per week of LPA, MPA, and VPA were determined as the time spent in 30–125 milligravities (mg), >125–400 mg, and >400 mg intensity activity, respectively[25]. The duration of MVPA was estimated as the sum of MPA and VPA.

### Outcomes ascertainment

The primary outcomes of interest in this study were all-cause, cancer, and CVD mortality. Information on the date and cause of death (C00-C97 for cancer and I00-I99 for CVD according to ICD-10) were obtained from death certificates held by the National Health Service (NHS) Information Center (England and Wales) and the NHS Central Register (Scotland). The follow-up time began at the accelerometer completion and ended at the occurrence of death or the end of follow-up (November 12, 2021), whichever came first.

### Covariates ascertainment

Covariates were selected based on a prior-defined directed acyclic graph (Supplementary Fig. 3). Our study finally included age (years between birth date and the start of accelerometry), sex (acquired from central registry at recruitment; Field ID: 31), ethnicity, education, smoking status, alcohol intake (product of intake frequency and gram of each type of alcohol per one standard drink)[33], diet quality, sleep quality, self-rated health status, history of cancer, CVD, hypertension, long-standing illness, disability, or infirmity, and experience of serious illness, injury, bereavement, stress in last two years derived from the questionnaires, body mass index (BMI, kg/m², weight in kilograms divided by height in meters squared) and waist circumstance (cm) measured during the initial visit, season and wear duration recorded by the accelerometers, and diabetes duration (years between the first occurrence of diabetes and the start of accelerometry). Diet scores (0–7) were by assigning 1 point for a healthy frequency and 0 point for

an unhealthy frequency of consumption of fruits, vegetables, fish, processed meat, unprocessed red meat, whole grains, and refined grains, with higher scores indicating a healthier diet quality[34,35]. Similarly, sleep scores (0–5) were generated by incorporating five sleep factors: chronotype, sleep duration, insomnia, snoring, and excessive daytime sleepiness, with higher scores indicating better sleep quality[36].

### Statistical analyses

Baseline characteristics of the participants according to different volumes of PA were described as means and standard deviations (SDs) for continuous variables and numbers (percentages) for categorical variables.

Dose–response associations of PA with all-cause and cause-specific mortality were evaluated using restricted cubic splines fitted in the Cox proportional hazards models ("rms" package in R). The reference values were set at the 1st percentile, and knots at the 5th, 35th, 65th, and 95th percentiles of the PA distribution. Potential non-linearity was tested by Wald tests. Then, the duration of PA was categorized into four levels: <1750, 1750–2099, 2100–2449, and ≥2450 min/week for LPA, <150, 150–299, 300–449, and ≥450 min/week for MPA, 0, 1–14, 15–29, and ≥30 min/week for VPA, and <275, 275–449, 450–624, and ≥625 min/week for MVPA after considering recommendations in the current PA guideline (≥150 min/week of MPA or ≥75 min/week of VPA), the inflection points of the dose–response relationship in this study (about 1750, 300, 30, and 275 min/week for LPA, MPA, VPA, and MVPA), and the sample size of each group. Cox proportional hazard models were performed to estimate the HRs and 95% CIs of all-cause and cause-specific mortality risk. Linear trends were examined by entering the median value of each category of PA as a continuous variable into the models. Three multivariable-adjusted models were constructed to account for potential confounding: model 1 was the crude model that adjusted for age (continuous, in years), sex (male or female), ethnicity (white or others), education (college/university or others), season (spring, summer, autumn, or winter) and duration of the accelerometry (continuous); model 2 was further adjusted for smoking status (never, former, or current), alcohol intake (continuous, in gram/day), diet scores (categories, 0–7), and sleep scores (continuous, 0–5); model 3 was the main model that additionally adjusted for BMI (continuous, in kg/m²), waist circumference (continuous, in cm), self-rated health (excellent, good, fair, or poor), long-standing illness, disability or infirmity (yes or no), illness, injury, bereavement, stress in last 2 years (yes or no), history of cancer or CVD (yes or no), history of hypertension (yes or no), and diabetes duration (continuous, in years). In the primary analyses, age, season at the time of accelerometry recording, and history of cancer or CVD were treated as time-varying covariates by constructing interaction terms between covariates and time. The proportional hazard assumption for Cox models was checked with Schoenfeld residuals and no violation was observed (Supplementary Table 27). Furthermore, a risk matrix was used to investigate the joint associations of different-intensity PA with all-cause and cause-specific mortality. The combination of the least active groups was set as the reference group. The PAFs were calculated to estimate the percentage of all death cases that would be prevented if individuals in the less active group were as active as the most active one[37,38]. Missing values in covariates with a missing rate <1% (e.g., smoking status and BMI) were completely excluded, while those with a missing rate >1% (e.g., education and diet scores) were coded as an additional category for categorical variables or mean values for continuous variables.

Stratified analyses were conducted according to age (<60 years and ≥60 years), sex (male and female), BMI (<25 kg/m² and ≥25 kg/m²), waist circumference (<102 cm for male and <88 cm for female, and other), smoking status (never and ever), alcohol intake (<28 g/day for male and <14 g/day for female, and other), diet scores (<4 and ≥4), sleep scores (<3 and ≥3) and history of hypertension (yes and no) to

examine whether the associations varied by these factors. Interaction terms were tested by a wald test.

Several sensitivity analyses were carried out to assess the robustness of our results. First, participants with cancer or CVD at baseline were excluded when assessing the association between PA and corresponding cause-specific mortality. Second, participants with poor self-rated health status were excluded, considering that they were more likely to undertake less PA and have higher risks of mortality. Third, we excluded participants who died within the first 2 and 4 years of follow-up to avoid the potential risk of reverse causation. Fourth, we performed stratified analyses by the number of diabetes severity factors, including glycated hemoglobin level (HbA1c) ≥ 7.0%, diabetes duration ≥10 years, and insulin medication use. Fifth, these factors were treated as covariates in the adjusted models. Sixth, T2D-related complications including related eye disease were additionally adjusted in the model. Seventh, LPA, MPA, and VPA variables were mutually adjusted to evaluate whether the associations were attributable to other intensity. Eighth, we stratified the cancer deaths by T2D/Obesity-related cancers and T2D/Obesity-independent cancers. Ninth, we repeated the main analysis after imputing the missing values of the covariates using chained equation multiple imputations (Supplementary eMethods). Tenth, we employed the Fine and Gray sub-distribution hazard model to examine the associations between PA and CVD mortality, accounting for cancer deaths as a competing risk. Similarly, we considered CVD deaths as a competing risk when analyzing the relationship between PA and cancer mortality through competing risk analyses. Finally, total PA in MET-min/week was estimated as the sum of three PA modes (LPA, MPA, and VPA) according to guidelines for data processing of the International Physical Activity Questionnaire[39]. Additionally, we repeated the analysis in all-cause mortality after stratified the participants into two distinct groups: those who meet the current guideline PA recommendations and those who did not align with the recommendations.

All the analyses were conducted using STATA 16 statistical software (Stata Corp LLP, College Station, TX) and R software (version 4.1.3). The statistical significance was set as $P < 0.05$ (two-sided test).

### Reporting summary

Further information on research design is available in the Nature Portfolio Reporting Summary linked to this article.

## Data availability

The main data used in this study were accessed from the publicly available UK Biobank Resource (https://www.ukbiobank.ac.uk) under application number 79095, which cannot be shared with other investigators due to data privacy laws. The UK Biobank data can be accessed by researchers on the application. The outcomes of analyses are included in the Source data file. Source data are provided with this paper.

## Code availability

Scripts used to perform the analyses are available at https://github.com/Chen-jie-Xu/UKB_accelerometer_PA_mortality_T2D.git.

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

## Acknowledgements

This work was supported by National Natural Science Foundation of China (No. 72204071 to C.X.); Zhejiang Provincial Natural Science Foundation of China (No. LY23G030005 to C.X.); Scientific Research Foundation for Scholars of HZNU (No. 4265C50221204119 to C.X.). This study was conducted using the UK Biobank resource (Application 79095). We want to express our sincere thanks to the participants of the UK Biobank, and the members of the survey, development, and management teams of this project. The icons in the top three boxes of Fig. 1 were derived from Iconfont (https://www.iconfont.cn). The icons in the exposure and outcomes boxes of Fig. 1 were designed by Amethyst Prime, Freepik, Kalashnyk, Mikan933, Photo3idea_studio from Flaticon (https://www.flaticon.com). We extend our sincere thanks to the designers at Iconfont and Flaticon.

## Author contributions

Z.C. and J.M. contributed to this work equally. C.X. and K.S. were involved in the conception, design, and conduct of the study and the analysis and interpretation of the results. Z.C. and J.M. wrote the first draft of the manuscript, and all authors edited, reviewed, and approved the final version of the manuscript. Z.C., J.M., and H.C. analyzed the data and interpreted the results. C.X., K.S., Y.H., and H.Y. have made critical revision of the manuscript for important intellectual content. C.X. had full access to all the data in the study and takes responsibility for the integrity of the data and the accuracy of the data analysis.

## Competing interests

The authors declare no competing interests.
