## [Peer Review File · Nature Communications]

REVIEWER COMMENTS

Reviewer #1 (Remarks to the Author):

This is a prospective cohort study aimed to investigate the association of device-measured PA with all-cause, cancer, and 15 cardiovascular disease (CVD) mortality among individuals with type 2 diabetes (T2D). The findings are expected and may advance the knowledge gap in this field. The strengths of this study include large sample size, objective measure on physical activity, and a reasonable follow-up duration. I have some comments for the consideration of the authors.

1. It is not well justified why shall we concern about physical activity specifically in patients with T2D. What is unique in T2D when compared with general populations.
2. How was the index day of follow-up defined? Was this the date of diagnosis with T2DM
3. It is unclear what kind of multiple imputations approach was used. Which variables were imputed? Which variables were included as predictors in the imputation model? How many imputations were performed? Were effects estimated using Rubin's rules?
4. As sleep quality is closely associated with physical activity, this parameter shall be also included into adjusted models. How sleep duration was defined in model 2? Using self-reported data or device-measured sleep duration?
5. It seems not make sense to group participants into 5 groups in table 2 as the number of cases was so small.
6. How was incidence rate per 1000 person-year when compared with other cohorts or large clinical trials?
7. Less than 50% of the causes of death were due to cardiovascular diseases. Did this include death of unknown origin as cardiovascular or not? Overall, 60-70% of the causes of death in people with T2D is expected to be cardiovascular. Please check and clarify.

Reviewer #2 (Remarks to the Author):

Review

The study titled "Association between device-measured physical activity and all-cause and cause-specific mortality among individuals with T2D" investigates the relationship between physical activity (PA) and mortality rates in individuals with type 2 diabetes (T2D) using the UK Biobank. The study uses accelerometer-measured PA data and finds that different intensities and durations of PA are associated with lower risks of all-cause and cause-specific mortality among individuals with T2D. More vigorous physical activity (VPA) was associated with larger risk reductions in mortality than moderate physical activity (MPA) and light physical activity (LPA).

Key findings include:

There were L-shaped associations with physical activity duration for all-cause and cancer mortality. This suggests that the risk of death decreases with increased physical activity up to a certain point, after which the risk plateaus.

Negative linear associations were observed for cardiovascular disease mortality, indicating that the risk of death from cardiovascular disease decreases steadily with increased physical activity.

Following current guideline recommendations could reduce the risk of all-cause mortality by 44% for moderate-intensity physical activity and by 69% for vigorous-intensity physical activity.

Strengths:

The study uses a large-scale sample size and rich data sources, which enabled accurate identification of patients with T2D.

The use of accelerometer-measured PA data is a strength as it captures both leisure and non-leisure PA across multiple domains, minimizing recall bias and reporting bias.

The study provides a comprehensive analysis of the relationship between PA and mortality in T2D patients, considering different intensities and durations of PA.

Weaknesses:

The study acknowledges that there is a lack of robust evidence on whether a 7-day measurement is representative of habitual PA. This could potentially underestimate the true association between PA and mortality.

The study acknowledges the possibility of reverse causation, as PA could be influenced by participants' undiagnosed diseases or health status.

The study is an observational cohort study, making it difficult to make causal conclusions. Despite adjusting for potentially relevant confounders, the possibility of unmeasured confounders that might bias the results cannot be rejected.

The UK Biobank, which was used for the study, had a low response rate, and participants are not representative of the overall UK population. This could potentially limit the generalizability of the findings.

Major Points:

1. The rates of cancer vs. CVD mortality reported in T2D in UK Biobank seem quite different from prior studies. This study (<https://jamanetwork.com/journals/jamainternalmedicine/fullarticle/2530287>) reports almost twice as many CVD deaths vs. cancer deaths. What explanation do the authors have for this? How representative is the UK biobank? Could there be competing interests? In other words, are people with T2D who sign up for UK Biobank and sign up to wear accelerometer (slightly) more active/healthy than their counterparts who are not as active, leading to less CVD in this group, and more cancers instead?

2. The authors also talk about modifiable risk factors; this appears to be in the same context as cancer as well as CVD. Not all cancers relate to diabetes and/or are preventable. Do the authors have the ability to stratify cancer deaths by T2D/Obesity-related cancers and T2D/Obesity-independent cancers?

3. In the introduction, I do not see discussion of other large studies in people with T2D and objectively measured physical activity, such as Look AHEAD and perhaps DPP/DPPOS (in the context of prediabetes). I do see some of these discussed in the discussion section. The authors should describe these studies in the context of their study and highlight what their study adds in terms of novelty that is not known from other studies on PA and mortality.

4. A total of 118,113 subjects in the UK biobank had T2D, but 22,041 had accelerometer data. This may not be reflective of the overall group with T2D. Have the authors done a sensitivity analysis, and what were the findings?

5. MPA vs VPA separation. How was this cutoff defined and why? A common assessment is MVPA combined, such as was done in a number of papers from the Look AHEAD study and other studies. Can the authors combine MPA and VPA into MVPA to confirm the findings?

6. The accelerometer is wrist-worn. Can the authors comment on what activity may/may not be captured by wrist-worn accelerometers vs. for example hip-worn accelerometers.
7. The MPA and VPA are reported in 'x mg intense activity'. Can these be converted/expressed in METs or at least compared to METs? How do their definitions of MPA, LPA, and VPA relate to categorization in other studies?
8. Generally, participants who were 'healthier' by self-report were more active. Have the authors corrected for this factor in the Models, which could limit one's physical activity?
9. In line 84-86 that authors mention that they reviewed characteristics of in- and excluded subjects, but don't comment on it. Please elaborate whether there were differences.
10. Are data on diabetes-related complications available? With longer-duration of T2DM, neuropathy and other complications can reduce the ability to be physically active.
11. The data are combined for male and female. Have the authors done an analysis by sex?
12. The cancer reduction seems greatest at the highest level of LPA, whereas the greatest reduction for CVD mortality is achieved by highest level of VPA. How can this be reconciled? Are causes of death competing, thereby introducing bias/survival bias?
13. In line 155, the authors state that guidelines direct >75/mins of VPA per week. This is not fully correct for ADA guidelines; these suggest that this level of activity is feasible for younger, more fit individuals, in addition to the overall recommendation of >150min of MVPA. The authors should adjust the statement, as this may not be feasible for all patients with T2D, in particular older, less fit patients.

Minor Points:

1. Line 201 PAFs, please explain abbreviation, if not done prior.

Reviewer #3 (Remarks to the Author):

This is a well written paper and a thorough analysis on the association of accelerometer measured physical activity with mortality in individuals with type 2 diabetes. I have a few comments to consider.

Line 173-175: The authors write "This may also explain why more T2D patients (94.0%) in our study met the minimum PA recommendation 174 (150 minutes/week of MPA) than those (23.8%) in the other study [14]."

There may be more reasons why the PA estimates are high and should be acknowledged, such as the 1) selection bias of the UK biobank which represents ~5% of the UK population and may bias toward healthier population and thus more active and 2) wrist worn accelerometry may overestimate activity in some individuals

It would be helpful to compare physical activities levels measured by accelerometer from other studies of populations with T2D or prediabetes. As one example, Look AHEAD trial has accelerometer measured physical activity that may be useful for comparison. Also comparing results from such studies focuses on T2D populations with accelerometer data and similar outcomes used in the current paper, like mortality, CVD incidence could be useful, such as: doi: 10.2337/dc21-1206.

Lines 178-184: Goes beyond the scope/purpose of this paper when discussing current guidelines. I would suggest removing or keeping specific to type 2 diabetes population.

In the main paper, I suggest specifically listing all the interactions you tested rather than generalizing this and leaving it to the reader to look in the supplement to learn which variables were tested. In other words update to list as... age, sex, BMI, WC, Smoking Status, Alcohol Intake, Diet Score, Sleep Duration, Hypertension....somewhere in the methods section.

Line 58 - should 'density' be 'intensity'?

The supplement tables are well done and thorough. I have very minor edits:

Supplement Table S7, S18, S19, S22 - it looks like the titles were cut off at the ends.

Supplement Figure S3. What does TDI stand for? Can you add this below the figure?

Reviewer #4 (Remarks to the Author):

This paper studies the association between accelerometer-derived physical activity with all-cause and cause-specific mortality among individuals with type 2 diabetes in UK Biobank study. I have some methodological and practical concerns. Please see the attached detailed comments to the Author.

Review

This paper studies the association between accelerometer-derived physical activity with all-cause and cause-specific mortality among individuals with type 2 diabetes in UK Biobank study. Overall the paper is well written. However, I have the following methodological and practical concerns which needs to be addressed.

Major Concerns

- The article’s purpose is not clear from the abstract and the introduction. It needs to be more direct. Association between objectively measured PA and all cause or Cancer/CVD mortality is not new. The novelty of this work needs to be pointed out in terms of existing literature.
- The abstract, introduction, results and discussion should be carefully revised to ensure that it does not suggest a causal association based on an observational study. Although this has been mentioned in the discussion, more careful rewording is necessary in terms of capturing the association and the estimated effect in previous sections. Given the statistical method (Cox model) followed in the paper, it is not right to make causal claims such as “If current guideline recommendations were followed, the risk of all-cause mortality might be reduced by..” or say “Assuming the associations to be causal, 26.0% of death cases in the study population were attributed to...”.
- There is a lack of literature review in the introduction in terms of existing work establishing relation between objectively measured PA and all cause, CVD or cancer mortality (e.g., Leroux et al. 2021 in UK Biobank, and several others).
- The paper investigates the relation between the duration of light intensity PA (LPA), MPA, and VPA assessed by accelerometers and risks of all-cause, cancer, and cardiovascular disease (CVD) mortality. However, other modalities of PA such as fragmentation metrics (ASTP, SATP), the diurnal pattern of PA, distributional pattern could be equally important, which needs to be acknowledged.
- In terms of population characteristics, comorbidities such as Heart attack, Stroke, High blood pressure and other key confounders such as illness or injury in the past 2 years, longstanding illness or disability, need to be adjusted for in the analysis.
- Line 99 “If current PA recommendations were followed, the risk of all-cause 100 mortality might be reduced by..”, can this be shown visually? Again causal claims should be avoided.

- In Statistical analysis section, Model 1 should be clearly written (if possible in forms of an equation), before extending to Model 2 and 3.
- “Dose-response associations of PA with all-cause and cause-specific mortality were evaluated using restricted cubic splines fitted in the Cox proportional hazards models”, more details need to be presented on the estimation method and the software that was used to fit the model, so that the analysis might be replicated.
- In terms of the statistical modelling, Cancer, CVD mortality are competing risks. In a competing risks regression framework, Fine-Gray model might be an alternative to separate cause specific models, which could be explored as an additional analysis offering important insights.

Minor Concerns

- In terms of assessing possible selection bias from missing data, supplementary table S4 shows the characteristics of participants with type 2 diabetes included and excluded from the current study. The findings from this table should be mentioned in the main paper.
- P might be replaced by ‘p-value’ throughout the paper. Importantly, The actual p-value needs to be reported in the main paper unless they are < 0.001 (very small).
- The proportional hazard assumption diagnostics needs to be presented in the supplementary material, at least for the main model.
- State (name) and refer to the testing method used for testing non linearity: “P for non-linearity < 0.05 ”
- “Then, the duration of PA was categorized into five levels mainly following the cut-off points defined by the current PA recommendations.” Be explicit above the cut-off points and/or cite relevant literature.
- Thorough proofreading is recommended.

****Responses to the Reviewer #1****

Comment 1: This is a prospective cohort study aimed to investigate the association of device-measured PA with all-cause, cancer, and 15 cardiovascular disease (CVD) mortality among individuals with type 2 diabetes (T2D). The findings are expected and may advance the knowledge gap in this field. The strengths of this study include large sample size, objective measure on physical activity, and a reasonable follow-up duration.

Response: We appreciate the reviewer's assessment and positive comments. According to your valuable suggestions, we have carefully revised the manuscript as below.

Comment 2: I have some comments for the consideration of the authors. It is not well justified why shall we concern about physical activity specifically in patients with T2D. What is unique in T2D when compared with general populations.

Response: Thank you for your important comment. We have rewritten the introduction section to justify the **uniqueness and necessity** of studying the association between PA and mortality risk specifically in T2D patients as follows.

[Introduction (page 3; lines 35-42)] *“The mortality risk in diabetic adults was approximately 60% higher than that in the non-diabetic (Pearson-Stuttard J, et al., Lancet Diabetes Endocrinol 2021). Physical activity (PA) has been reported as an indispensable factor in lowering or even cancelling out this excess mortality risk. However, more physical inactivity was observed in diabetic adults (Abildso CG, et al., MMWR Morb Mortal Wkly Rep 2023; Centers for Disease Control and Prevention (2022). Preventing Complications of Diabetes: Statistics Report), partly because they may face more physiological and psychological barriers to PA (Nesti L, et al., Cardiovasc Diabetol 2019; Zahalka SJ, et al., The Role of Exercise in Diabetes 2023). Their complicated conditions may also bias the benefit from PA (Shin WY, et al., J Korean Med Sci 2018). Investigating the association between PA and mortality specifically in diabetic adults is vital to inform PA recommendations tailored to this high-risk population.”*

Comment 3: How was the index day of follow-up defined? Was this the date of diagnosis with T2DM.

Response: We thank the reviewer for pointing out this issue. In our study, the follow-up began on the date when the 7-day accelerometry ended, because physical activity was our exposure factor. Participants were diagnosed with T2D before the start of the follow-up. Duration of prevalent T2D (from diagnosis to baseline of this study) has been included into the adjusted model.

Comment 4: It is unclear what kind of multiple imputations approach was used. Which variables were imputed? Which variables were included as predictors in the imputation model? How many imputations were performed? Were effects estimated using Rubin's rules?

Response: We apologize for the lack of details about multiple imputation, which have been added to the **supplementary material** as follows:

[Supplementary Text 1. Details about multiple imputation]:

Multiple imputation

We used multiple imputations by chained equations to impute missing values for the following covariates: alcohol intake, smoking status, diet scores, sleep scores, self-rated health status, long-standing illness, disability or infirmity, illness, injury, bereavement, stress in last 2 years. The imputations model included all of covariates as predictors. We performed 10 imputations and pooled all results across the imputed datasets using Rubin's rules to obtain final effect estimates.

Missing proportion of covariates needed to impute

Alcohol intake (missing proportion: 0.09%), smoking status (0.31%), diet scores (5.21%), sleep scores (0.06%), self-rated health status (0.31%), long-standing illness, disability or infirmity (1.88%), illness, injury, bereavement, stress in last 2 years (2.86%).

Comment 5: As sleep quality is closely associated with physical activity, this parameter shall be also included into adjusted models. How sleep duration was defined in model 2? Using self-

reported data or device-measured sleep duration?

Response: In the previous version, sleep duration, based on self-reported data, was defined as the number of hours participants slept in a 24-hour period, including naps. Your comments are very enlightening to us. Not only the sleep duration, but also the sleep quality is closely related to physical activity. Thus, in the revised manuscript, we have replaced the sleep duration with a sleep score that captures five aspects of sleep, including chronotype, sleep duration, insomnia, snoring, and excessive daytime sleepiness (Li X, et al., *J Am Coll Cardiol* 2021; Huang BH, et al., *Br J Sports Med* 2022; Cao Z, et al., *J Affect Disord* 2023). The revised manuscript was listed as follows:

[Methods (page 17; lines 316-318)] *“Similarly, sleep scores (0 to 5) were generated by incorporating five sleep factors: chronotype, sleep duration, insomnia, snoring, and excessive daytime sleepiness, with higher scores indicating better sleep quality (Cao Z, et al., J Affect Disord 2023).”*

Comment 6: It seems not make sense to group participants into 5 groups in table 2 as the number of cases was so small.

Response: Thanks for the professional suggestion. We fully agree that grouping participants into five groups may have resulted in an imbalance in the number of cases across groups and affected the statistical power. Therefore, we regrouped the participants into four groups after taking into account the physical activity guideline, the inflection points of the dose-response relationship in this study, and the sample size of each group. All analyses have been revised accordingly.

[Methods (page 17; lines 328-335)] *“Then, the duration of PA was categorized into four levels: <1750, 1750-2099, 2100-2449, and ≥2450 minutes/week for LPA, <150, 150-299, 300-449, and ≥450 minutes/week for MPA, 0, 1-14, 15-29, and ≥30 minutes/week for VPA, and <275, 275-449, 450-624, and ≥625 minutes/week for MVPA after considering recommendations in the current PA guideline (≥150 minutes/week of MPA or ≥75 minutes/week of VPA), the inflection points of the dose-response relationship in this study (about 1750, 400, 15, and 375*

minutes/week for LPA, MPA, VPA, and MVPA), and the sample size of each group.”

Comment 7: How was incidence rate per 1000 person-year when compared with other cohorts or large clinical trials?

Response: We thank the reviewer for this comment. As can be seen in the following table, compared with other large cohorts, the mortality rates in T2D patients in our study were relatively low.

Cohort	Country	Age, years	Year	N	Mortality rate (per 1000 person years)		
					All-cause	Cancer	CVD
UK Biobank (the current study)	England, Scotland, and Wales	62.3 (SD: 7.8)	2013-2015 (baseline), 2021 (end)	19 624	6.0	3.3	1.2
Clinical Practice Research Datalinkas (Pearson-Stuttard J, et al., Lancet Diabetes Endocrinol 2021)	England	M: 67 (57-77) F: 66 (53-79)	2018	313 907	M: 27.8 F: 29.5	M: 9.3 F: 8.1	M: 7.5 F: 7.1
National Health Interview Survey (Gregg EW, et al., Lancet 2018)	United States	60.6 (SE: 0.2)	2010-2014	21.1 million	15.2	3.0	5.2
China Kadoorie Biobank (Bragg F, et al., JAMA 2017)	China	57.2 (SD: 10.1)	2004-2008 (baseline), 2014 (end)	512 869	13.7	3.0	5.4
Diabetes Care	Auckland & New	56.7 (SD: 13.8)	1994-2018	45 072	9.9	3.5	2.0

Support Service (Yu D, et al., Lancet Glob Health 2021)	Zealand						
---	---------	--	--	--	--	--	--

Abbreviations: M, male; F, female; SD, standard deviation; SE, standard error; CVD, cardiovascular disease.

As most of cohort studies, including the well-known Framingham Heart Study and the Health Professionals Follow-up Study, the UK Biobank was not representative of the general population. Its response rate was only 5.5% though the sample size was as large as 0.5 million. The lower mortality rates in the UK Biobank participants were partly attributed to their more favorable risk factor profile. A prior study reported that UK Biobank participants were less likely to be obese, to smoke, and to drink alcohol and had fewer self-reported health conditions (Fey A, et al., *Am J Epidemiol* 2017). A noteworthy point is that the “healthy volunteer” selection bias in the UK Biobank may affect the generalizability (external validity), but it was demonstrated less likely to distort the association between physical activity and mortality (internal validity). Stamatakis et al. used poststratification to match the UK Biobank sample to the general UK population (the Health Survey for England 2008) in terms of sociodemographic characteristics and prevalence of lifestyle risk factors (physical inactivity, alcohol intake, smoking, and poor diet) and found that the hazard ratios with and without poststratification were closely aligned across all-cause, cancer, and CVD mortality as the following table shows (Stamatakis E, et al., *Epidemiology* 2021).

TABLE 2. Adjusted RHR^a of Each Lifestyle Risk Factor for All-cause, CVD, and Cancer Mortality, Excluding People with History of Cancer or CVD (n = 302,009 After Exclusion)

Variable	Level	RHRs (Poststratified/Unweighted)		
		All-cause Mortality	CVD Mortality	Cancer Mortality
Physical activity level	≥7.5 MET hours/week	Reference	Reference	Reference
	>0, <7.5 MET hours/week	1.04 (0.99, 1.08)	0.92 (0.82, 1.03)	1.06 (0.99, 1.19)
	No PA	1.01 (0.90, 1.13)	0.83 (0.57, 1.13)	1.21 (0.98, 1.45)

Above table was citing from Stamatakis E, et al., *Epidemiology* 2021.

a. RHR, ratio of hazard ratio.

To clarify this, we have revised the manuscript as follows:

[Discussion (page 13; lines 249-256)] *“Sixth, though the sample size of the UK Biobank was as large as 0.5 million, the response rate was only 5.5% and accelerometer-measured PA was only available for a subset of participants. Diabetic adults in these analyses were generally healthier, more physically active, and had a lower mortality rate than the overall diabetic adults. The selection bias may limit the generalizability of the current findings. Nonetheless, recent evidence suggests that any potential bias resulting from poor representativeness would have a minimal impact on the association of PA with mortality in the UK Biobank (Stamatakis E, et al., Epidemiology 2021).”*

Comment 8: Less than 50% of the causes of death were due to cardiovascular diseases. Did this include death of unknown origin as cardiovascular or not? Overall, 60-70% of the causes of death in people with T2D is expected to be cardiovascular. Please check and clarify.

Response: Thanks very much for this insightful comment. We confirm that deaths of unknown origin were not included as cardiovascular deaths in our analysis after checking the data carefully.

According to data from 0.3 million individuals with diabetes in the Clinical Practice Research Datalink, a well-known primary care database in England, **improvements in cancer death rates were lagging behind the vast improvements in vascular disease death rates, resulting in a transition from vascular causes to cancers as the leading contributor to death rates in individuals with diagnosed diabetes** (Pearson-Stuttard J, et al., *Lancet Diabetes Endocrinol* 2021). The proportion of deaths due to vascular diseases declined from 44% in 2001 to 24% in 2018 while that due to cancer increased from 22% in 2001 to 28% in 2018. Such proportions were not consistent with ours (CVD: 21.2%; cancer: 54.4%) because **diabetic individuals in the UK Biobank were not representative of those derived from the general UK population**. However, as we replied in Comment 7, **the unrepresentativeness may not distort the association between physical activity and mortality** (internal validity) though it may affect the generalizability (external validity).

We have revised the manuscript as follows:

[Discussion (page 13; lines 249-256)] *“Sixth, though the sample size of the UK Biobank was as large as 0.5 million, the response rate was only 5.5% and accelerometer-measured PA was only available for a subset of participants. Diabetic adults in these analyses were generally healthier, more physically active, and had a lower mortality rate than the overall diabetic adults. The selection bias may limit the generalizability of the current findings. Nonetheless, recent evidence suggests that any potential bias resulting from poor representativeness would have a minimal impact on the association of PA with mortality in the UK Biobank (Stamatakis E, et al., Epidemiology 2021).”*

****Responses to the Reviewer #2****

Comment 1: The study titled "Association between device-measured physical activity and all-cause and cause-specific mortality among individuals with T2D" investigates the relationship between physical activity (PA) and mortality rates in individuals with type 2 diabetes (T2D) using the UK Biobank. The study uses accelerometer-measured PA data and finds that different intensities and durations of PA are associated with lower risks of all-cause and cause-specific mortality among individuals with T2D. More vigorous physical activity (VPA) was associated with larger risk reductions in mortality than moderate physical activity (MPA) and light physical activity (LPA).

Key findings include:

There were L-shaped associations with physical activity duration for all-cause and cancer mortality. This suggests that the risk of death decreases with increased physical activity up to a certain point, after which the risk plateaus.

Negative linear associations were observed for cardiovascular disease mortality, indicating that the risk of death from cardiovascular disease decreases steadily with increased physical activity. Following current guideline recommendations could reduce the risk of all-cause mortality by 44% for moderate-intensity physical activity and by 69% for vigorous-intensity physical activity.

Strengths:

The study uses a large-scale sample size and rich data sources, which enabled accurate identification of patients with T2D.

The use of accelerometer-measured PA data is a strength as it captures both leisure and non-leisure PA across multiple domains, minimizing recall bias and reporting bias.

The study provides a comprehensive analysis of the relationship between PA and mortality in T2D patients, considering different intensities and durations of PA.

Weaknesses:

The study acknowledges that there is a lack of robust evidence on whether a 7-day measurement is representative of habitual PA. This could potentially underestimate the true

association between PA and mortality.

The study acknowledges the possibility of reverse causation, as PA could be influenced by participants' undiagnosed diseases or health status.

The study is an observational cohort study, making it difficult to make causal conclusions. Despite adjusting for potentially relevant confounders, the possibility of unmeasured confounders that might bias the results cannot be rejected

The UK Biobank, which was used for the study, had a low response rate, and participants are not representative of the overall UK population. This could potentially limit the generalizability of the findings.

Response: We thank the reviewer for the insightful review of our study and appreciate the thoughtful evaluation of the strengths and weaknesses.

Comment 2: Major Points: The rates of cancer vs. CVD mortality reported in T2D in UK Biobank seem quite different from prior studies. This study (<https://jamanetwork.com/journals/jamainternalmedicine/fullarticle/2530287>) reports almost twice as many CVD deaths vs. cancer deaths. What explanation do the authors have for this? How representative is the UK biobank? Could there be competing interests? In other words, are people with T2D who sign up for UK Biobank and sign up to wear accelerometer (slightly) more active/healthy than their counterparts who are not as active, leading to less CVD in this group, and more cancers instead?

Response: Thank you for your careful and professional review. We have listed the mortality rates of diabetic adults in some prior large cohorts in the following table. As you point out, the rates of cancer vs. CVD mortality in our study does differ from some prior studies.

Cohort	Country	Age, years	Year	N	Mortality rate (per 1000 person years)		
					All-cause	Cancer	CVD
UK Biobank (the current study)	England, Scotland, and	62.3 (7.8)	2013-2015 (baseline), 2021	19 624	6.0	3.3	1.2

	Wales		(end)				
Clinical Practice Research Datalink (Pearson-Stuttard J, et al., Lancet Diabetes Endocrinol 2021)	England	M: 67 (57-77) F: 66 (53-79)	2018	313 907	M: 27.8 F: 29.5	M: 9.3 F: 8.1	M: 7.5 F: 7.1
National Health Interview Survey (Gregg EW, et al., Lancet 2018)	United States	60.6 (SE: 0.2)	2010-2014	21.1 million	15.2	3.0	5.2

Abbreviations: M, male; F, female; SE, standard error; CVD, cardiovascular disease.

Few incidence rates from cohort studies were representative of the general population, unless they relied on national population-based registry. The response rate of the UK Biobank was only 5.5% though its sample size was as large as 0.5 million. As you supposed, **the risk factor profile and mortality rates were more favorable in the UK Biobank.** A prior study reported that UK participants were less likely to be obese, to smoke, and to drink alcohol and had fewer self-reported health conditions (Fry A, et al., *Am J Epidemiol* 2017). Therefore, neither the absolute magnitude of the mortality rates nor the relative magnitude of CVD deaths vs cancer deaths in this study were representative and **they cannot be directly compared with other studies, which may also be unrepresentative.**

A noteworthy point is that **the “healthy volunteer” selection bias in the UK Biobank may affect the generalizability (external validity), but it was demonstrated less likely to distort the association between physical activity and mortality (internal validity).** Stamatakis et al. used poststratification to match the UK Biobank sample to the general UK population (the Health Survey for England 2008) in terms of sociodemographic characteristics and prevalence of lifestyle risk factors (physical inactivity, alcohol intake, smoking, and poor diet) and found that the hazard ratios with and without poststratification were closely aligned across all-cause, cancer, and CVD mortality as the following table shows (Stamatakis E, et al.,

Epidemiology 2021).

TABLE 2. Adjusted RHR^a of Each Lifestyle Risk Factor for All-cause, CVD, and Cancer Mortality, Excluding People with History of Cancer or CVD (n = 302,009 After Exclusion)

Variable	Level	RHRs (Poststratified/Unweighted)		
		All-cause Mortality	CVD Mortality	Cancer Mortality
Physical activity level	≥7.5 MET hours/week	Reference	Reference	Reference
	>0, <7.5 MET hours/week	1.04 (0.99, 1.08)	0.92 (0.82, 1.03)	1.06 (0.99, 1.19)
	No PA	1.01 (0.90, 1.13)	0.83 (0.57, 1.13)	1.21 (0.98, 1.45)

a. RHR, ratio of hazard ratio.

To clarify this, we have revised the manuscript as follows:

[Discussion (page 13; lines 249-256)] *“Sixth, though the sample size of the UK Biobank was as large as 0.5 million, the response rate was only 5.5% and accelerometer-measured PA was only available for a subset of participants. Diabetic adults in these analyses were generally healthier, more physically active, and had a lower mortality rate than the overall diabetic adults. The selection bias may limit the generalizability of the current findings. Nonetheless, recent evidence suggests that any potential bias resulting from poor representativeness would have a minimal impact on the association of PA with mortality in the UK Biobank (Stamatakis E, et al., Epidemiology 2021).”*

Comment 3: The authors also talk about modifiable risk factors; this appears to be in the same context as cancer as well as CVD. Not all cancers relate to diabetes and/or are preventable. Do the authors have the ability to stratify cancer deaths by T2D/Obesity-related cancers and T2D/Obesity-independent cancers?

Response: According to the reviewer’s suggestions, in the revised manuscript, we further stratified cancer deaths by T2D/Obesity-related cancers and T2D/Obesity-independent cancers (**Supplementary Table S26**). Both of their risks can be reduced by increased physical activity, though the associations between physical activity and T2D/Obesity-related cancer mortality were non-significant in many categories because the sample size was too small, which affected the statistical power.

We revised the manuscript as follows:

[Methods (page 20; lines 376-377)] *“Eighth, we stratified the cancer deaths by*

T2D/Obesity-related cancers and T2D/Obesity-independent cancers.”

[Results (page 8; lines 147-152)] *“The associations were slightly attenuated in some categories when we excluded participants who had cancer or CVD at baseline (Supplementary Table S24), when we excluded participants who died during the first 2 and 4 years of follow-up (Supplementary Table S25), or when cancer deaths were stratified into T2D/Obesity-related cancers or -independent cancers (Supplementary Table S26).”*

Table S26. Association of physical activity with T2D/Obesity-related or -independent cancer mortality risk (n = 19,624) *

Outcomes	LPA (minutes/week)			
	<1750	1750-2099	2100-2449	≥2450
T2D/Obesity-related cancer mortality				
No. of cases	29	25	16	14
HR (95% CI)	1 (ref)	0.91 (0.53,1.57)	0.75 (0.40,1.41)	1.06 (0.54,2.08)
T2D/Obesity-independent cancer mortality				
No. of cases	139	104	68	46
HR (95% CI)	1 (ref)	0.79 (0.61,1.03)	0.64 (0.48,0.87)	0.69 (0.48,0.97)
	MPA (minutes/week)			
	<150	150-299	300-449	≥450
T2D/Obesity-related cancer mortality				
No. of cases	9	25	24	26
HR (95% CI)	1 (ref)	1.11 (0.51,2.42)	1.04 (0.46,2.35)	0.88 (0.38,2.06)
T2D/Obesity-independent cancer				

mortality				
No. of cases	63	103	96	95
HR (95% CI)	1 (ref)	0.64 (0.46,0.89)	0.58 (0.41,0.82)	0.43 (0.30,0.63)
VPA (minutes/week)				
	0	1-14	15-29	≥30
T2D/Obesity-related cancer mortality				
No. of cases	23	31	20	10
HR (95% CI)	1 (ref)	1.09 (0.63,1.91)	0.87 (0.46,1.64)	0.62 (0.27,1.39)
T2D/Obesity-independent cancer mortality				
No. of cases	126	94	82	55
HR (95% CI)	1 (ref)	0.63 (0.48,0.84)	0.68 (0.51,0.92)	0.65 (0.46,0.93)
MVPA (minutes/week)				
	<275	275-449	450-624	≥625
T2D/Obesity-related cancer mortality				
No. of cases	31	22	19	12
HR (95% CI)	1 (ref)	0.72 (0.41,1.27)	0.84 (0.45,1.56)	0.68 (0.32,1.45)
T2D/Obesity-independent cancer mortality				
No. of cases	152	89	74	42
HR (95% CI)	1 (ref)	0.61 (0.46,0.80)	0.67 (0.50,0.91)	0.48 (0.33,0.70)

HR indicates hazard ratio; CI, confidence interval; LPA, light-intensity physical activity; MPA, moderate-intensity physical activity; VPA, vigorous-intensity physical activity; MVPA, moderate-to-vigorous-intensity physical activity.

* Hazard ratios were calculated in Cox proportional hazards model after adjusting for age (years), sex (male or female), ethnicity (white or others), education (college/university or others), season at the time of accelerometry recording (spring, summer, autumn, or winter), accelerometer wear duration (days), smoking status (never, former, or current), alcohol intake (g/day), diet score (0 to 7), sleep score (0 to 5), body mass index (kg/m²), waist circumference (cm), self-rated health (excellent, good, fair, or poor), long-standing illness, disability or infirmity (yes or no), illness, injury, bereavement, or stress in last 2 years (yes or no), history of cancer or cardiovascular disease (yes or no), history of hypertension (yes or no), and diabetes duration (years).

Comment 4: In the introduction, I do not see discussion of other large studies in people with T2D and objectively measured physical activity, such as Look AHEAD and perhaps DPP/DPPOS (in the context of prediabetes). I do see some of these discussed in the discussion section. The authors should describe these studies in the context of their study and highlight what their study adds in terms of novelty that is not known from other studies on PA and mortality.

Response: We thank the reviewer for this important comment. As suggested, we have added some evidence from previous relevant studies in the **Introduction**. We highlight how our study extends these studies by using wrist-worn accelerometers, which are more sensitive to activities involving upper body movement, in a much larger prospective cohort; by performing a dose-response analysis that can inform specific physical activity recommendations; and by examining the potential effect modifications by factors related to diabetes severity.

[Introduction (page 4; lines 57-70)] *“Accelerometry-based evidence in diabetic adults was scant. Two studies derived from the National Health and Nutrition Examination Survey (NHANES) and a study from the Look AHEAD Trial all demonstrated significant risk reductions in outcomes related to all-cause mortality with higher volumes of PA (Loprinzi PD, *Physiol Behav* 2016; Zhu P, et al., *Int J Behav Nutr Phys Act* 2023; Look AHEAD Study Group, *Diabetes Care* 2022). However, their sample sizes were less than 2000; accelerometers were worn on the hip or waist, which may omit activities that primarily involve upper body movement; and dose-response analyses were not performed.*

To fill the knowledge gaps, we aimed to investigate the dose-response associations between the duration of PA in different intensities and risks of all-cause, cancer, and cardiovascular disease (CVD) mortality among adults with type 2 diabetes (T2D; 96% of all diabetes cases (GBD 2021 Diabetes Collaborators, Lancet 2023)) in the UK Biobank, the largest prospective cohort with PA measured by wrist-worn accelerometers to date. Furthermore, we investigated the potential effect modification by factors related to diabetes severity, such as glycemic control, diabetes duration, and diabetes medication use.”

Comment 5: A total of 118,113 subjects in the UK biobank had T2D, but 22,041 had accelerometer data. This may not be reflective of the overall group with T2D. Have the authors done a sensitivity analysis, and what were the findings?

Response: We thank the reviewer for the insightful recommendation. As suggested, we performed a sensitivity analysis comparing the baseline characteristics between the included and the excluded diabetic adults (**Supplementary Table S4**). Results showed that **diabetic adults included in the analyses were generally healthier**. They had a higher level of education, a lower waist circumference, a lower prevalence of smoking, and better self-rated health status. **We have acknowledged the potential selection bias in the limitation section.**

We revised the manuscript as follows:

[Results (page 5; lines 79-82)] *“Diabetic participants included in the analyses were generally healthier than those who were excluded, with a higher level of education, a smaller body mass index (BMI) and waist circumference, a lower prevalence of smoking, and better health status at baseline (Supplementary Table S1).”*

[Discussion (page 13; lines 251-256)] *“Diabetic adults in these analyses were generally healthier, more physically active, and had a lower mortality rate than the overall diabetic adults. The selection bias may limit the generalizability of the current findings. Nonetheless, recent evidence suggests that any potential bias resulting from poor representativeness would have a minimal impact on the association of PA with mortality in the UK Biobank (Stamatakis E, et al., Epidemiology 2021).”*

Table S4. Characteristics of participants with type 2 diabetes included in or excluded from the current study

Characteristics	Excluded*	Included
Total, n	98,757	19,624
Age, year, mean (SD)	61.64 (8.04)	62.26 (7.79)
Sex, male, n (%)	46,511 (47.10)	8,713 (44.40)
Ethnicity, white, n (%)	88,988 (91.10)	18761 (95.60)
Education, college or university, n (%)	27,030 (28.91)	8,057 (42.29)
BMI, kg/m², mean (SD)	28.46 (5.42)	27.34 (5.04)
Waist, cm, mean (SD)	93.27 (14.76)	90.13 (14.24)
Smoking status, n (%)		
Never	51,654 (52.91)	10,910 (55.60)
Former	34,649 (35.49)	7,315 (37.28)
Current	11,322 (11.60)	1,399 (7.13)
Diet score, mean (SD)	3.57 (1.51)	3.70 (1.49)
Sleep score, mean (SD)	2.92 (1.03)	3.06 (0.99)
Alcohol intake, g/day, mean (SD)	16.68 (21.25)	17.89 (19.91)
Diabetes duration, year, mean (SD)	5.75 (2.01)	5.84 (2.06)
HbA1c, mean (SD)	41.66 (13.07)	38.99 (10.46)
Insulin medication use, n (%)	4,714 (4.77)	564 (2.87)
Self-rated health, n (%)		
Excellent	12,247 (12.58)	3,730 (19.01)
Good	51,632 (53.02)	11,393 (58.06)
Fair	25,809 (26.50)	3,690 (18.80)
poor	7,697 (7.90)	811 (4.13)
History of cancer or CVD, n (%)	20,616 (20.88)	3,294 (16.79)
History of hypertension, n (%)	33,913 (34.34)	5,312 (27.07)

Long-standing illness, disability or infirmity, n (%)	41,184 (43.08)	6,854 (35.56)
Illness, injury, bereavement, stress in last 2 years, n (%)	44,851 (47.77)	8,379 (43.92)

* Participants with missing information or poor-quality information on wear time, quality of wear, daylight saving crossover, smoking status, alcohol intake, BMI, ethnicity, waist, sleep score, and self-rated health at baseline were excluded.

BMI indicates body mass index; CVD, cardiovascular disease; SD, standard deviation.

Comment 6: MPA vs VPA separation. How was this cutoff defined and why? A common assessment is MVPA combined, such as was done in a number of papers from the Look AHEAD study and other studies. Can the authors combine MPA and VPA into MVPA to confirm the findings?

Response: We thank the reviewer for this advice. In our study, MPA and VPA were separated by a cut-off of 125 milligravities used in previous studies (Petermann-Rocha F, et al. *BMC Med* 2021; Ho FK, et al. *BMC Med* 2022).

As the reviewer suggested, in the revised manuscript, we have combined MPA and VPA into MVPA and added the corresponding results to the main text. Overall, compared with the lowest category of MVPA, a higher duration of MVPA was associated with a lower risk of all-cause and cause-specific mortality, which is consistent with the primary conclusion of our study.

We revised the manuscript as follows:

[Methods (page 15; line 292)] “The duration of MVPA was estimated as the sum of MPA and VPA.”

Table 2. Hazard ratios (95% CI) for all-cause mortality according to physical activity among individuals with type 2 diabetes

Exposures	No. of cases	Incidence rate per 1000	Hazard ratio (95% CI) *
-----------	--------------	-------------------------	-------------------------

	person-year		Model 1	Model 2	Model 3
LPA (minutes/week)					
<1750	336	9.58	1 (Ref)	1 (Ref)	1 (Ref)
1750-2099	229	5.62	0.66 (0.56,0.78)	0.68 (0.58,0.81)	0.79 (0.66,0.93)
2100-2449	149	4.27	0.53 (0.44,0.65)	0.55 (0.45,0.67)	0.66 (0.54,0.80)
≥2450	96	4.02	0.54 (0.43,0.68)	0.56 (0.44,0.71)	0.70 (0.55,0.89)
P for trend			<0.001	<0.001	<0.001
MPA (minutes/week)					
<150	182	23.56	1 (Ref)	1 (Ref)	1 (Ref)
150-299	255	9.40	0.47 (0.39,0.57)	0.50 (0.41,0.60)	0.59 (0.48,0.72)
300-449	190	5.20	0.31 (0.26,0.39)	0.34 (0.27,0.42)	0.44 (0.36,0.55)
≥450	183	2.90	0.22 (0.18,0.27)	0.24 (0.19,0.30)	0.35 (0.27,0.44)
P for trend			<0.001	<0.001	<0.001
VPA (minutes/week)					
0	322	13.36	1 (Ref)	1 (Ref)	1 (Ref)
1-14	229	5.90	0.52 (0.44,0.62)	0.54 (0.46,0.64)	0.63 (0.53,0.75)
15-29	172	4.61	0.45 (0.37,0.54)	0.47 (0.39,0.57)	0.60 (0.50,0.74)
≥30	87	2.53	0.30 (0.24,0.38)	0.33 (0.26,0.42)	0.46 (0.36,0.60)
P for trend			<0.001	<0.001	<0.001
MVPA (minutes/week)					
<275	394	14.27	1 (Ref)	1 (Ref)	1 (Ref)
275-449	202	5.33	0.46 (0.39,0.55)	0.49 (0.41,0.58)	0.57 (0.48,0.68)
450-624	134	3.92	0.40 (0.33,0.49)	0.42 (0.35,0.52)	0.54 (0.44,0.66)
≥625	80	2.29	0.29 (0.22,0.37)	0.31 (0.24,0.40)	0.43 (0.33,0.55)
P for trend			<0.001	<0.001	<0.001

HR indicates hazard ratio; CI, confidence interval; LPA, light-intensity physical activity; MPA, moderate-intensity physical activity; VPA, vigorous-intensity physical activity; MVPA, moderate-to-vigorous-intensity physical activity.

* Hazard ratios (95% CI) were calculated in Cox proportional hazards model: model 1, adjusted for age (years), sex (male or female), ethnicity (white or others), education (college/university or others), season at the time of accelerometry recording (spring,

summer, autumn, or winter), and accelerometer wear duration (days); model 2, further adjusted for smoking status (never, former, or current), alcohol intake (g/day), diet score (0 to 7), and sleep score (0 to 5) based on model 1; model 3, further adjusted for body mass index (kg/m²), waist circumference (cm), self-rated health (excellent, good, fair, or poor), long-standing illness, disability or infirmity (yes or no), illness, injury, bereavement, or stress in last 2 years (yes or no), history of cancer or cardiovascular disease (yes or no), history of hypertension (yes or no), and diabetes duration (years) based on model 2.

Comment 7: The accelerometer is wrist-worn. Can the authors comment on what activity may/may not be captured by wrist-worn accelerometers vs. for example hip-worn accelerometers.

Response: Thanks for the valuable comment! Accelerometers can be worn at various sites of the body (e.g., dominant/non-dominant wrist, chest, waist, hip, thigh, ankle), which are sensitive to different types of physical activity and no single-site-worn accelerometers can fully capture the context of all physical activities. For example, accelerometers at the thigh or the ankle can quantify cycling that others cannot, while those at the wrist can capture activities with predominant arm movements, such as writing, cooking, and folding laundry, which are likely underestimated by accelerometers at the other sites. Though no consensus has been reached on the optimal wear-site, there has been a marked shift from hip/waist to wrist placement in some large epidemiological studies because wrist-worn accelerometers are perceived as less intrusive, with a longer wear time, a better compliance, an improved capability of assessing sleep behaviors, and a capacity of capturing more non-ambulatory arm movements in daily living (Gao Z, et al., *J Clin Med* 2021; Duncan MJ, et al., *Eur J Sport Sci* 2020). In the current study, we focused on overall activity volumes and intensities rather than specific activities. In addition, the accuracy of discrimination between intensities (inactive, light, moderate, vigorous) was higher for the dominant wrist (at least 75.0%) than the waist (at least 60.5%) (Mielke GI, et al., *Scand J Med Sci Sports* 2023). Therefore, wrist accelerometry is appropriate for our study. To clarify this, we have added a statement to the

Limitations section as follows:

[Discussion (page 13; lines 238-241)] *“Second, we utilized wrist-worn accelerometers that are more accurate in discriminating different intensities compared to waist-worn accelerometers (Mielke, G. I, et al., Scand J Med Sci Sports 2023). However, they may not fully capture activities involving minimal arm movement, such as cycling, that ankle-worn accelerometers can quantify more accurately.”*

Comment 8: The MPA and VPA are reported in ‘x mg intense activity’. Can these be converted/expressed in METs or at least compared to METs? How do their definitions of MPA, LPA, and VPA relate to categorization in other studies?

Response: We thank the reviewer for these insightful comments. Physical activity duration data have been converted into metabolic equivalent of task (MET) units using the standard criteria consistent with the International Physical Activity Questionnaire (IPAQ): LPA MET-minutes/week = 3.3*LPA minutes/week; MPA MET-minutes/week = 4.0*MPA minutes/week; VPA MET-minutes/week = 8.0*VPA minutes/week. Total physical activity in MET-minutes/week were calculated as the sum of LPA, MPA, and VPA. As noted by the reviewer, expressing activity levels in METs may facilitate comparisons with other studies. Results based on MET were similar to the original ones, demonstrating the robustness of our conclusion (**Supplementary Table S22; Supplementary Figure S3**). Revisions were as follows:

[Methods (page 20; lines 381-383)] *“Finally, total PA in MET-minutes/week was estimated as the sum of three PA modes (LPA, MPA, and VPA) according to guidelines for data processing of the International Physical Activity Questionnaire (IPAQ) – Short and Long Forms. 2005).”*

[Results (page 8; lines 137-145)] *“The results were largely consistent with the main analyses when ... when the volume of PA was expressed in metabolic of equivalent task (MET) (Supplementary Table S22 and Figure S3).”*

Table S22. Association of physical activity in 600 MET with all-cause and cause-specific mortality risk

Exposures	HR (95% CI) of mortality per 600 MET-minutes/week increment *		
	All-cause	Cancer	Cardiovascular disease
Total physical activity	0.92 (0.90,0.94)	0.93 (0.90,0.96)	0.94 (0.90,0.98)

HR indicates hazard ratio; CI, confidence interval; LPA, light-intensity physical activity; MPA, moderate-intensity physical activity; VPA, vigorous-intensity physical activity; MVPA, moderate-to-vigorous-intensity physical activity; MET, metabolic equivalent of task.

* Hazard ratios were calculated per 600 MET-minutes/week increment of total physical activity in Cox proportional hazards model after adjusting for age (years), sex (male or female), ethnicity (white or others), education (college/university or others), season at the time of accelerometry recording (spring, summer, autumn, or winter), accelerometer wear duration (days), smoking status (never, former, or current), alcohol intake (g/day), diet score (0 to 7), sleep score (0 to 5), body mass index (kg/m²), waist circumference (cm), self-rated health (excellent, good, fair, or poor), long-standing illness, disability or infirmity (yes or no), illness, injury, bereavement, or stress in last 2 years (yes or no), history of cancer or cardiovascular disease (yes or no), history of hypertension (yes or no), and diabetes duration (years).

Figure S3. Dose-response association of total physical activity in MET with all-cause and cause-specific mortality

HR indicates hazard ratio; CI, confidence interval; LPA, light-intensity physical activity; MPA, moderate-intensity physical activity; VPA, vigorous-intensity physical activity; MVPA, moderate-to-vigorous-intensity physical activity.

All adjusted for age (years), sex (male or female), ethnicity (white or others), education (college/university or others), season at the time of accelerometry recording (spring, summer, autumn, or winter), accelerometer wear

duration (days), smoking status (never, former, or current), alcohol intake (g/day), diet score (0 to 7), sleep score (0 to 5), body mass index (kg/m²), waist circumference (cm), self-rated health (excellent, good, fair, or poor), long-standing illness, disability or infirmity (yes or no), illness, injury, bereavement, or stress in last 2 years (yes or no), history of cancer or cardiovascular disease (yes or no), history of hypertension (yes or no), and diabetes duration (years).

Different cut points have been developed to discriminate accelerometer-measured LPA, MPA, and VPA, such as the activity count-based, the step-based, and the gravitational unit-based cut points (Gao Z, et al., *J Clin Med* 2021). However, no consensus has been reached about which approach is the best or what the most appropriate cut points are. In addition, the company-specific, proprietary algorithms make it difficult to compare the data from different studies using different accelerometers. In this study, we used the gravitational unit-based cut points derived from a study published on the *Circulation* (Ho FK, et al., *Circulation* 2022). Comparisons with other studies, either using accelerometer-measured or self-reported PA, either in T2D patients or in the general population, along with the potential reasons underlying the between-study differences, have been thoroughly discussed in the Discussion section.

Comment 9: Generally, participants who were ‘healthier’ by self-report were more active. Have the authors corrected for this factor in the Models, which could limit one’s physical activity?

Response: We thank the reviewer for pointing out this issue. As suggested, we have incorporated self-rated health status (*excellent, good, fair, or poor*) as a covariate in the fully-adjusted models in the revised version. Results remained mostly consistent with the original analyses.

Comment 10: In line 84-86 that authors mention that they reviewed characteristics of in- and excluded subjects, but don’t comment on it. Please elaborate whether there were differences.

Response: Thank you for your suggestion. We have elaborated the differences between the in-

and excluded subjects in the revised manuscript. Diabetic adults included in the analyses were generally healthier than those who were excluded. They had a higher level of education, a lower waist circumference, a lower prevalence of smoking, and better self-rated health status. Accordingly, we acknowledged in the limitation section that this may have led to some degree of selection bias.

[Results (page 5; lines 79-82)] *“Diabetic participants included in the analyses were generally healthier than those who were excluded, with a higher level of education, a smaller body mass index (BMI) and waist circumference, a lower prevalence of smoking, and better health status at baseline (Supplementary Table S1).”*

[Discussion (page 13; lines 251-256)] *“Diabetic adults in these analyses were generally healthier, more physically active, and had a lower mortality rate than the overall diabetic adults. The selection bias may limit the generalizability of the current findings. Nonetheless, recent evidence suggests that any potential bias resulting from poor representativeness would have a minimal impact on the association of PA with mortality in the UK Biobank (Stamatakis E, et al., Epidemiology 2021).”*

Comment 11: Are data on diabetes-related complications available? With longer-duration of T2DM, neuropathy and other complications can reduce the ability to be physically active.

Response: Thank you for your suggestion. As suggested, we have performed a sensitivity analysis that additionally adjusted for diabetes-related complications, including diabetes-related eye disease. The results of this analysis are shown in **Supplementary Table S18** and were consistent with the primary results, demonstrating the robustness of our conclusion. Revisions were listed below:

[Methods (page 19; line 373-374)] *“Sixth, T2D-related complications including related eye disease were additionally adjusted in the model.”*

[Results (page 8; line 137-141)] *“The results were largely consistent with the main analyses ... when we additionally adjusted for T2D-related complications (Supplementary Table S18).”*

Comment 12: The data are combined for male and female. Have the authors done an analysis by sex?

Response: Thank you for your reminder. In the original version, we have done a subgroup analysis by sex and found no significant interactions between any intensity of physical activity and sex, except for LPA (Supplementary Tables S11-14). In the revised manuscript, we have clarified what subgroup analyses were performed and the corresponding results:

[Methods (page 19; lines 358-359)] “Stratified analyses were conducted according to age (<60 years and ≥60 years), sex (male and female) ...”

[Results (page 7; lines 133-137)] “No statistically significant interactions were found between all PA intensities and age, BMI, waist circumference, smoking status, alcohol intake, diet scores, sleep scores, and history of hypertension, except sex. (p-value for interaction > 0.05, Supplementary Tables S11-14). The association between LPA and all-cause mortality was stronger among female than male (p-value for interaction = 0.031).”

For your convenience, results of the subgroup analyses by sex were picked out and merged into one table as follows.

Association between PA and the risk of all-cause mortality by sex

Sex	Duration of PA (minutes/week)				p-value for interaction
	<1750	1750-2099	2100-2449	≥2450	
LPA					
Female	1 (ref)	0.56 (0.41,0.76)	0.64 (0.47,0.87)	0.64 (0.46,0.90)	0.031
Male	1 (ref)	0.93 (0.75,1.14)	0.64 (0.49,0.84)	0.69 (0.48,0.98)	
MPA					
	<150	150-299	300-449	≥450	
Female	1 (ref)	0.66 (0.47,0.92)	0.44 (0.30,0.64)	0.39 (0.27,0.58)	0.719
Male	1 (ref)	0.57 (0.45,0.73)	0.47 (0.36,0.62)	0.33 (0.24,0.45)	
VPA					

	0	1-14	15-29	≥30	
Female	1 (ref)	0.57 (0.43,0.77)	0.69 (0.50,0.95)	0.57 (0.37,0.86)	0.291
Male	1 (ref)	0.68 (0.55,0.85)	0.57 (0.45,0.74)	0.41 (0.29,0.57)	
MVPA					
	<275	275-449	450-624	≥625	
Female	1 (ref)	0.49 (0.36,0.66)	0.53 (0.38,0.74)	0.51 (0.35,0.74)	0.135
Male	1 (ref)	0.64 (0.51,0.79)	0.55 (0.42,0.72)	0.35 (0.24,0.52)	

HR indicates hazard ratio; CI, confidence interval; LPA, light-intensity physical activity; MPA, moderate-intensity physical activity; VPA, vigorous-intensity physical activity; MVPA, moderate-to-vigorous-intensity physical activity.

* Hazard ratios were calculated in Cox proportional hazards model after adjusting for age (years), sex (male or female), ethnicity (white or others), education (college/university or others), season at the time of accelerometry recording (spring, summer, autumn, or winter), accelerometer wear duration (days), smoking status (never, former, or current), alcohol intake (g/day), diet score (0 to 7), sleep score (0 to 5), body mass index (kg/m²), waist circumference (cm), self-rated health (excellent, good, fair, or poor), long-standing illness, disability or infirmity (yes or no), illness, injury, bereavement, or stress in last 2 years (yes or no), history of cancer or cardiovascular disease (yes or no), history of hypertension (yes or no), and diabetes duration (years).

Comment 13: The cancer reduction seems greatest at the highest level of LPA, whereas the greatest reduction for CVD mortality is achieved by highest level of VPA. How can this be reconciled? Are causes of death competing, thereby introducing bias/survival bias?

Response: We appreciate the reviewer for the professional comment. In response to the thoughtful comments from you and other reviewers, we have repeated all analyses after categorizing physical activity into four groups and adjusting for more covariates such as self-reported health status in the revised version. In our revised analysis, compared with the lowest active group, the highest level of MPA was associated with a 52% lower risk of cancer mortality whereas the highest level of VPA was associated with a 69% lower risk of CVD mortality (Figure 2). In response to your first query about the discrepancy between the risk reduction of

cancer and CVD mortality in relation to MPA and VPA, **we would like to clarify that the METs (Metabolic Equivalent of Task) in the MPA and VPA groups are different for their reference groups and maximal activity groups.** Consequently, it would be inappropriate to directly compare HRs of MPA and VPA maximal activity groups for cancer and CVD mortality. To reconcile this issue, we have referred to a study (Wang Y, et al., *JAMA Intern Med* 2021) where the proportion of VPA to MVPA was computed. We explored the association of the proportion of VPA to MVPA with cancer and CVD mortality, and compared the HR values for both outcomes at equal VPA proportions. **The results, as reflected in the following table, show no significant difference in HR values between cancer mortality and CVD mortality within the same group of VPA proportion.** Therefore, it suggested that MPA and VPA confer similar benefits for both outcomes among the diabetic population.

In addition, as the reviewer noted, there may be competing risks between deaths from cancer and CVD, which may cause survival bias. Therefore, **we conducted additional sensitivity analysis to control for these potential sources of bias:** Firstly, we used a Fine-Gray regression model to account for competing risks. This allows us to accurately estimate the cause-specific hazard of death in the presence of competing risks. Secondly, regarding survival bias, we have conducted a sensitivity analysis excluding individuals who died within the first two or four years of follow-up. This approach minimizes the potential influence of pre-existing conditions that could lead to early death and distort the observed associations. The results of these analyses, presented in the **Supplementary Tables S21 and S25**, were largely consistent with our main findings, further illustrating the robustness of our conclusions. The revised manuscript was listed below:

[Methods (page 19; line 369-370)] *“Third, we excluded participants who died within the first 2 and 4 years of follow-up to avoid the potential risk of reverse causation.”*

[Methods (page 20; line 379-381)] *“Tenth, we used the Fine and Gray subdistribution hazard model to examine the associations between PA and mortality, incorporating CVD death or cancer death as a competing risk.”*

[Results (page 8; line 137-150)] *“The results were largely consistent with the main*

analyses when... when the competing risk regression model was used (Supplementary Table S21)... when we excluded participants who died during the first 2 and 4 years of follow-up (Supplementary Table S25)...

Associations of the proportion of VPA to MVPA with all-cause and cause-specific mortality among participants who had any MVPA.

	Proportion of VPA to MVPA, HR (95% CI)				
	0%	>0 % to 3%	>3% to 4.6%	>4.6% to 7.4%	>7.4%
Mortality	(n = 3,621)	(n = 4,156)	(n = 3,818)	(n = 4,111)	(n = 3,915)
All-cause	1.00 (Ref)	0.57 (0.46,0.71)	0.61 (0.49,0.75)	0.67 (0.54,0.82)	0.54 (0.43,0.70)
Cancer- cause	1.00 (Ref)	0.67 (0.50,0.90)	0.70 (0.53,0.94)	0.79 (0.60,1.04)	0.64 (0.46,0.89)
CVD-cause	1.00 (Ref)	0.69 (0.43,1.11)	0.66 (0.41,1.04)	0.80 (0.52,1.23)	0.49 (0.28,0.85)

HR indicates hazard ratio; CI, confidence interval; LPA, light-intensity physical activity; MPA, moderate-intensity physical activity; VPA, vigorous-intensity physical activity; MVPA, moderate-vigorous-intensity physical activity.

* Hazard ratios were calculated in Cox proportional hazards model after adjusting for age (years), sex (female or male), ethnicity (white or other), education (college/university or other), season at the time of accelerometry recording (spring, summer, autumn, or winter), accelerometer wear duration (days), smoking status (never, former, or current), alcohol intake (g/day), diet score (0 to 7), sleep score (0 to 5), body mass index (kg/m²), waist circumference (cm), self-rated health (excellent, good, fair, or poor), long-standing illness, disability or infirmity (yes or no), illness, injury, bereavement, stress in last 2 years (yes or no), history of cancer or cardiovascular disease (yes or no), history of hypertension (yes or no), and diabetes duration (years).

Figure 2. Association between accelerometer-derived physical activity and all-cause and cause-specific mortality among participants with type 2 diabetes.

All-cause mortality:

Cancer mortality:

Cardiovascular disease mortality:

HR indicates hazard ratio; CI, confidence interval; LPA, light-intensity physical activity; MPA, moderate-intensity physical activity; VPA, vigorous-intensity physical activity; MVPA, moderate-to-vigorous-intensity physical activity.

All adjusted for age (years), sex (male or female), ethnicity (white or others), education (college/university or others), season at the time of accelerometry recording (spring, summer, autumn, or winter), accelerometer wear duration (days), smoking status (never, former, or current), alcohol intake (g/day), diet score (0 to 7), sleep score (0 to 5), body mass index (kg/m²), waist circumference (cm), self-rated health (excellent, good, fair, or poor), long-standing illness, disability or infirmity (yes or no), illness, injury, bereavement, or stress in last 2 years (yes or no), history of cancer or cardiovascular disease (yes or no), history of hypertension (yes or no), and diabetes duration (years).

Table S21. Association of physical activity with cancer and cardiovascular disease mortality risk using Fine & Gray models for competing risk (n = 19,624) *

Outcomes	LPA (minutes/week)			
	<1750	1750-2099	2100-2449	≥2450
Cancer mortality				
HR (95% CI)	1 (ref)	0.82 (0.64,1.03)	0.67 (0.51,0.87)	0.75 (0.55,1.02)
Cardiovascular disease mortality				
HR (95% CI)	1 (ref)	0.81 (0.55,1.19)	0.90 (0.60,1.35)	0.78 (0.45,1.35)
	MPA (minutes/week)			
	<150	150-299	300-449	≥450
Cancer mortality				
HR (95% CI)	1 (ref)	0.70 (0.52,0.95)	0.64 (0.47,0.88)	0.49 (0.35,0.69)
Cardiovascular disease mortality				
HR (95% CI)	1 (ref)	0.89 (0.58,1.38)	0.55 (0.33,0.91)	0.53 (0.31,0.90)
	VPA (minutes/week)			
	0	1-14	15-29	≥30
Cancer mortality				
HR (95% CI)	1 (ref)	0.71 (0.56,0.91)	0.72 (0.55,0.93)	0.65 (0.47,0.91)
Cardiovascular disease mortality				
HR (95% CI)	1 (ref)	0.73 (0.50,1.07)	0.77 (0.52,1.16)	0.32 (0.16,0.62)
	MVPA (minutes/week)			
	<275	275-449	450-624	≥625
Cancer mortality				
HR (95% CI)	1 (ref)	0.63 (0.50,0.81)	0.71 (0.54,0.93)	0.52 (0.37,0.72)
Cardiovascular disease mortality				
HR (95% CI)	1 (ref)	0.58 (0.40,0.85)	0.58 (0.37,0.90)	0.31 (0.16,0.61)

HR indicates hazard ratio; CI, confidence interval; LPA, light-intensity physical activity; MPA, moderate-intensity physical activity; VPA, vigorous-intensity physical activity; MVPA, moderate-to-vigorous-intensity physical activity.

* Hazard ratios were calculated in Cox proportional hazards model after adjusting for age (years), sex (male or female), ethnicity (white or others), education (college/university or others), season at the time of accelerometry recording (spring, summer, autumn, or winter), accelerometer wear duration (days), smoking status (never, former, or current), alcohol intake (g/day), diet score (0 to 7), sleep score (0 to 5), body mass index (kg/m²), waist circumference (cm), self-rated health (excellent, good, fair, or poor), long-standing illness, disability or infirmity (yes or no), illness, injury, bereavement, or stress in last 2 years (yes or no), history of cancer or cardiovascular disease (yes or no), history of hypertension (yes or no), and diabetes duration (years).

Table S25. Association of physical activity with all-cause and cause-specific mortality risk, excluding deaths that occurred within the first 2 or 4 years of follow-up*

	All-cause mortality	Cancer mortality	Cardiovascular disease mortality
Excluding deaths that occurred within the first 2 years of follow-up (n = 19,476)			
LPA (minutes/week)			
<1750	1 (ref)	1 (ref)	1 (ref)
1750-2099	0.83 (0.68,1.00)	0.85 (0.65,1.10)	0.92 (0.60,1.40)
2100-2449	0.73 (0.59,0.91)	0.74 (0.55,1.00)	1.04 (0.66,1.63)
≥2450	0.75 (0.58,0.97)	0.84 (0.60,1.18)	0.72 (0.37,1.37)
MPA (minutes/week)			
<150	1 (ref)	1 (ref)	1 (ref)
150-299	0.63 (0.50,0.79)	0.90 (0.63,1.30)	0.85 (0.52,1.40)
300-449	0.51 (0.40,0.66)	0.86 (0.59,1.25)	0.58 (0.33,1.02)
≥450	0.40 (0.31,0.53)	0.67 (0.45,1.00)	0.58 (0.32,1.05)
VPA (minutes/week)			
0	1 (ref)	1 (ref)	1 (ref)

0-14	0.69 (0.57,0.84)	0.80 (0.60,1.05)	0.80 (0.53,1.23)
15-29	0.68 (0.55,0.84)	0.84 (0.62,1.13)	0.84 (0.53,1.34)
≥30	0.52 (0.39,0.69)	0.77 (0.54,1.10)	0.34 (0.16,0.73)
MVPA (minutes/week)			
<275	1 (ref)	1 (ref)	1 (ref)
275-449	0.61 (0.50,0.74)	0.66 (0.50,0.87)	0.62 (0.40,0.95)
450-624	0.62 (0.49,0.78)	0.82 (0.61,1.10)	0.65 (0.39,1.08)
≥625	0.46 (0.35,0.62)	0.57 (0.40,0.83)	0.37 (0.18,0.77)
Excluding deaths that occurred within the first 4 years of follow-up (n = 19,263)			
LPA (minutes/week)			
<1750	1 (ref)	1 (ref)	1 (ref)
1750-2099	0.92 (0.73,1.16)	0.96 (0.70,1.33)	0.89 (0.53,1.49)
2100-2449	0.80 (0.62,1.05)	0.79 (0.54,1.14)	0.99 (0.57,1.72)
≥2450	0.86 (0.63,1.18)	1.03 (0.69,1.54)	0.68 (0.31,1.50)
MPA (minutes/week)			
<150	1 (ref)	1 (ref)	1 (ref)
150-299	0.75 (0.57,0.99)	1.40 (0.85,2.29)	0.68 (0.38,1.20)
300-449	0.59 (0.43,0.80)	1.27 (0.76,2.13)	0.47 (0.24,0.91)
≥450	0.50 (0.36,0.69)	1.06 (0.62,1.81)	0.46 (0.23,0.93)
VPA (minutes/week)			
0	1 (ref)	1 (ref)	1 (ref)
0-14	0.82 (0.65,1.03)	0.92 (0.65,1.28)	0.97 (0.59,1.61)
15-29	0.74 (0.57,0.97)	0.93 (0.64,1.34)	0.81 (0.45,1.47)
≥30	0.49 (0.34,0.71)	0.66 (0.42,1.06)	0.32 (0.12,0.87)
MVPA (minutes/week)			
<275	1 (ref)	1 (ref)	1 (ref)
275-449	0.66 (0.52,0.84)	0.76 (0.54,1.05)	0.54 (0.32,0.92)
450-624	0.62 (0.47,0.82)	0.78 (0.54,1.14)	0.59 (0.31,1.09)

≥ 625	0.53 (0.38,0.75)	0.67 (0.43,1.05)	0.33 (0.13,0.82)
------------	------------------	------------------	------------------

HR indicates hazard ratio; CI, confidence interval; LPA, light-intensity physical activity; MPA, moderate-intensity physical activity; VPA, vigorous-intensity physical activity; MVPA, moderate-to-vigorous-intensity physical activity.

* Hazard ratios were calculated in Cox proportional hazards model after adjusting for age (years), sex (male or female), ethnicity (white or others), education (college/university or others), season at the time of accelerometry recording (spring, summer, autumn, or winter), accelerometer wear duration (days), smoking status (never, former, or current), alcohol intake (g/day), diet score (0 to 7), sleep score (0 to 5), body mass index (kg/m^2), waist circumference (cm), self-rated health (excellent, good, fair, or poor), long-standing illness, disability or infirmity (yes or no), illness, injury, bereavement, or stress in last 2 years (yes or no), history of cancer or cardiovascular disease (yes or no), history of hypertension (yes or no), and diabetes duration (years).

Comment 14: In line 155, the authors state that guidelines direct >75 /mins of VPA per week. This is not fully correct for ADA guidelines; these suggest that this level of activity is feasible for younger, more fit individuals, in addition to the overall recommendation of >150 min of MVPA. The authors should adjust the statement, as this may not be feasible for all patients with T2D, in particular older, less fit patients.

Response: Thank you for your insightful comment. Following the suggestion, we have removed the previous sentence which may cause ambiguity for readers, and revised the manuscript to present the results of PAFs objectively. The revised manuscript is listed below:

[Discussion (page 9; line 158-159)] *“As high as 38.6% of death could be prevented if T2D patients performed ≥ 625 minutes/week of MVPA.”*

[Discussion (page 11; line 195-196)] *“The latest guideline recommends ≥ 150 minutes/week of MPA to most T2D patients or ≥ 75 minutes/week of VPA to patients who are younger and more physically fit (ElSayed, N. A. et al., Diabetes Care 2023).”*

Minor Points:

Comment 15: 1. Line 201 PAFs, please explain abbreviation, if not done prior.

Response: As suggested, PAFs have been explained in the revised manuscript as follows:

[**Methods (page 19; lines 358-359)**] *“For example, the Population-attributable fractions (PAFs) for mortality sequentially decreased with higher durations of MVPA, with PAFs of 28.9%, 6.3%, and 3.4% for those engaging <275 minutes/week, 275-450 minutes/week, and 450-624 minutes/week, respectively.”*

****Responses to the Reviewer #3****

Comment 1: This is a well written paper and a thorough analysis on the association of accelerometer measured physical activity with mortality in individuals with type 2 diabetes.

Response: We appreciate the reviewer's positive comments. Your affirmation is valuable to us and reinforces the significance of our efforts and work. We have carefully considered your comments and worked to improve our study in the revised manuscript. If you have any further valuable suggestions or comments on our study, we would be delighted to hear them.

Comment 2: I have a few comments to consider. Line 173-175: The authors write "This may also explain why more T2D patients (94.0%) in our study met the minimum PA recommendation (150 minutes/week of MPA) than those (23.8%) in the other study [14]."

There may be more reasons why the PA estimates are high and should be acknowledged, such as the 1) selection bias of the UK biobank which represents ~5% of the UK population and may bias toward healthier population and thus more active and 2) wrist worn accelerometry may overestimate activity in some individuals

Response: Thanks very much for the insightful comments. We fully agree with the reviewer that the selection bias in the UK biobank participants (healthier and more active) and the potential measurement error in wrist-worn accelerometry may also attribute to the relatively high estimates of physical activity in our study. In this version of manuscript, we have removed the comparison of PA levels in our study with others in the original section, because that section mainly focused on comparisons of magnitude of associations between PA and mortality. We have acknowledged the selection bias and measurement error in wrist-worn accelerometry in the limitation section instead.

[Discussion (page 13; lines 238-241)] *"Second, we utilized wrist-worn accelerometers that are more accurate in discriminating different intensities compared to waist-worn accelerometers (Mielke GI, et al., Scand J Med Sci Sports 2023). However, they may not fully capture activities involving minimal arm movement, such as cycling, that ankle-worn accelerometers can quantify more accurately."*

[Discussion (page 13; lines 249-256)] *“Sixth, though the sample size of the UK Biobank was as large as 0.5 million, the response rate was only 5.5% and accelerometer-measured PA was only available for a subset of participants. Diabetic adults in these analyses were generally healthier, more physically active, and had a lower mortality rate than the overall diabetic adults. The selection bias may limit the generalizability of the current findings. Nonetheless, recent evidence suggests that any potential bias resulting from poor representativeness would have a minimal impact on the association of PA with mortality in the UK Biobank (Stamatakis E, et al., Epidemiology 2021).”*

Comment 3: It would be helpful to compare physical activities levels measured by accelerometer from other studies of populations with T2D or prediabetes. As one example, Look AHEAD trial has accelerometer measured physical activity that may be useful for comparison. Also comparing results from such studies focuses on T2D populations with accelerometer data and similar outcomes used in the current paper, like mortality, CVD incidence could be useful, such as: doi: 10.2337/dc21-1206.

Response: We thank the reviewer for the valuable advice. We agree that it would be helpful to compare physical activity levels and the magnitude of associations between other accelerometry-based studies and ours. The pity is that such evidence was scant. We only found three accelerometry-based studies conducted specifically in T2D populations. None of their physical activity levels can be directly compared with ours. Two studies were derived from the NHANES--one focused on daily ambulatory movement without quantifying the intensity, while the other focused on the benefits from replacing sedentary time with physical activity. The third study was derived from the Look AHEAD Trial, in which participants received tailored physical activity intervention, therefore its PA level cannot be compared with ours in free-living conditions.

As for associations between PA and mortality in these studies, we have added related results in **lines 161-168** as follows: *“Prior accelerometry-based studies specific to T2D patients were scant but all demonstrated significant risk reductions in mortality-related*

outcomes with higher volumes of PA, such as a 29% lower risk of premature all-cause mortality for every 60-minute increase in daily ambulatory movement in the NHANES and a 2.6% lower risk of a composite CVD outcome including all-cause mortality for every 100 MET-minutes/week increase in MVPA from baseline to four years in the Look AHEAD Trial (Loprinzi PD, Physiol Behav 2016; Zhu P, et al., Int J Behav Nutr Phys Act 2023; Look AHEAD Study Group, Diabetes Care 2022). However, the magnitude of these associations cannot be directly compared with ours because the outcomes were slightly different and the NHANES did not account for PA intensity.”

Comment 4: Lines 178-184: Goes beyond the scope/purpose of this paper when discussing current guidelines. I would suggest removing or keeping specific to type 2 diabetes population.

Response: We thank the reviewer for pointing out this issue. As suggested, we have removed this part and kept specific to type 2 diabetes population.

Comment 5: In the main paper, I suggest specifically listing all the interactions you tested rather than generalizing this and leaving it to the reader to look in the supplement to learn which variables were tested. In other words update to list as... age, sex, BMI, WC, Smoking Status, Alcohol Intake, Diet Score, Sleep Duration, Hypertension... somewhere in the methods section.

Response: We thank the reviewer for pointing out this issue. As suggested, we have explicitly listed out all factors tested for interaction in both the Methods and Results sections as follows:

[Methods (page 19; lines 358-364)] *“Stratified analyses were conducted according to age (<60 years and ≥60 years), sex (male and female), BMI (<25 kg/m² and ≥25 kg/m²), waist circumference (<102 cm for male and <88 cm for female, and other), smoking status (never and ever), alcohol intake (<28 g/day for male and <14g/day for female, and other), diet scores (<4 and ≥4), sleep scores (<3 and ≥3) and history of hypertension (yes and no) to examine whether the associations varied by these factors. Interactions were tested by a likelihood-ratio test comparing models with and without product terms between PA and these factors.”*

[Results (page 7; lines 133-137)] *“No statistically significant interactions were found between all PA intensities and age, BMI, waist circumference, smoking status, alcohol intake, diet scores, sleep scores, and history of hypertension, except sex. (p-value for interaction > 0.05, Supplementary Tables S11-14). The association between LPA and all-cause mortality was stronger among female than male (p-value for interaction = 0.031).”*

Comment 6: Line 58 - should 'density' be 'intensity'?

Response: We apologize for the typo error. As suggested, “density” has been changed to “intensity”.

Comment 7: The supplement tables are well done and thorough. I have very minor edits: Supplement Table S7, S18, S19, S22 - it looks like the titles were cut off at the ends.

Response: Thanks for your careful review. We have revised the titles for these tables in the Supplementary material.

Comment 8: Supplement Figure S3. What does TDI stand for? Can you add this below the figure?

Response: Thanks for your advice. We have added “*TDI, Townsend Deprivation Index*” below the figure as suggested.

****Responses to the Reviewer #4****

Comment 1: This paper studies the association between accelerometer-derived physical activity with all-cause and cause-specific mortality among individuals with type 2 diabetes in UK Biobank study. I have some methodological and practical concerns. Please see the attached detailed comments to the Author.

Major Concerns

The article's purpose is not clear from the abstract and the introduction. It needs to be more direct. Association between objectively measured PA and all cause or Cancer/ CVD mortality is not new. The novelty of this work needs to be pointed out in terms of existing literature.

Response: We thank the reviewer for this valuable comment. There have been many studies on objectively measured PA and all-cause or cause-specific mortality in the general population, yet few in patients with diabetes. As suggested, we have thoroughly revised the Introduction by first justifying the necessity of investigating this association specifically in diabetic patients, then providing evidence from existing literature and their limitations, and finally highlighting the novelty of our study.

[Abstract (page 2; lines 17-21)] *“Physical activity (PA) has been recognized as an important factor for the survival of individuals with type 2 diabetes (T2D) by many studies using self-reported PA, which is subject to recall bias. However, evidence based on objectively measured PA was scant. We aimed to investigate the association of accelerometer-measured PA with all-cause, cancer, and cardiovascular disease (CVD) mortality among individuals with T2D.”*

[Introduction (page 3; lines 35-70)] *“The mortality risk in diabetic adults was approximately 60% higher than that in the non-diabetic (Pearson-Stuttard J, et al., Lancet Diabetes Endocrinol 2021). Physical activity (PA) has been reported as an indispensable factor in lowering or even cancelling out this excess mortality risk. However, more physical inactivity was observed in diabetic adults (Abildso CG, et al., MMWR Morb Mortal Wkly Rep 2023; Centers for Disease Control and Prevention (2022). Preventing Complications of Diabetes: Statistics Report), partly because they may face more physiological and psychological barriers*

to PA (Nesti L, et al., Cardiovasc Diabetol 2019; Zahalka SJ, et al., The Role of Exercise in Diabetes 2023). Their complicated conditions may also bias the benefit from PA (Shin WY, et al., J Korean Med Sci 2018). Investigating the association between PA and mortality specifically in diabetic adults is vital to inform PA recommendations tailored to this high-risk population... Accelerometry-based evidence in diabetic adults was scant. Two studies derived from the National Health and Nutrition Examination Survey (NHANES) and a study from the Look AHEAD Trial all demonstrated significant risk reductions in outcomes related to all-cause mortality with higher volumes of PA (Loprinzi PD, Physiol Behav 2016; Zhu P, et al., Int J Behav Nutr Phys Act 2023; Look AHEAD Study Group, Diabetes Care 2022). However, their sample sizes were less than 2000; accelerometers were worn on the hip or waist, which may omit activities that primarily involve upper body movement; and dose-response analyses were not performed.

To fill the knowledge gaps, we aimed to investigate the dose-response associations between the duration of PA in different intensities and risks of all-cause, cancer, and cardiovascular disease (CVD) mortality among adults with type 2 diabetes (T2D; 96% of all diabetes cases (GBD 2021 Diabetes Collaborators, Lancet 2023)) in the UK Biobank, the largest prospective cohort with PA measured by wrist-worn accelerometers to date. Furthermore, we investigated the potential effect modification by factors related to diabetes severity, such as glycemic control, diabetes duration, and diabetes medication use.”

Comment 2: The abstract, introduction, results and discussion should be carefully revised to ensure that it does not suggest a causal association based on an observational study. Although this has been mentioned in the discussion, more careful rewording is necessary in terms of capturing the association and the estimated effect in previous sections. Given the statistical method (Cox model) followed in the paper, it is not right to make causal claims such as “If current guideline recommendations were followed, the risk of all-cause mortality might be reduced by..” or say “Assuming the associations to be causal, 26.0% of death cases in the study population were attributed to...”.

Response: We thank the reviewer for pointing out this issue. As suggested, we have thoroughly checked the manuscript to ensure that all causal claims have been removed. In addition, in the limitation section, we emphasized that *“Fourth, as for any observational studies, we cannot exclude the role of residual/unmeasured confounding or make causal conclusions, even though we have adjusted for a wide range of potential confounders”* in **lines 250-252**.

Comment 3: There is a lack of literature review in the introduction in terms of existing work establishing relation between objectively measured PA and all cause, CVD or cancer mortality (e.g., Leroux et al. 2021 in UK Biobank, and several others).

Response: We thank the reviewer for pointing out this issue. As suggested, we have expanded the Introduction by including evidence from existing literature on objectively measured PA and mortality both in general population and in diabetic patients. Evidence in the general population indicates that the magnitude of association based on accelerometer-measured PA is significantly larger than that based on self-reported PA, while evidence in the diabetic patients can give us a picture about what have been found and the knowledge gaps.

[Introduction (page 4; lines 54-63)] *“A harmonized meta-analysis of eight studies found that in the general population the maximal risk reduction in all-cause mortality for accelerometer-measured MVPA (about 60%) was about twice the magnitude as reported in studies relying on self-reported PA (Ekelund U, et al., BMJ 2019). Accelerometry-based evidence in diabetic adults was scant. Two studies derived from the National Health and Nutrition Examination Survey (NHANES) and a study from the Look AHEAD Trial all demonstrated significant risk reductions in outcomes related to all-cause mortality with higher volumes of PA (Loprinzi PD, Physiol Behav 2016; Zhu P, et al., Int J Behav Nutr Phys Act 2023; Look AHEAD Study Group, Diabetes Care 2022). However, their sample sizes were less than 2000; accelerometers were worn on the hip or waist, which may omit activities that primarily involve upper body movement; and dose-response analyses were not performed.”*

Comment 4: The paper investigates the relation between the duration of light intensity PA

(LPA), MPA, and VPA assessed by accelerometers and risks of all-cause, cancer, and cardiovascular disease (CVD) mortality. However, other modalities of PA such as fragmentation metrics (ASTP, SATP), the diurnal pattern of PA, distributional pattern could be equally important, which needs to be acknowledged.

Response: We thank the reviewer for bringing this concern to our attention. As suggested, we have acknowledged this in the limitation part as follows:

[Discussion (page 13; lines 241-244)] *“Third, we only focused on the duration and intensity of PA, but the other properties of PA, such as fragmentation metrics, diurnal pattern, and distributional pattern may be equally important, which warrant further investigation.”*

Comment 5: In terms of population characteristics, comorbidities such as Heart attack, Stroke, High blood pressure and other key confounders such as illness or injury in the past 2 years, longstanding illness or disability, need to be adjusted for in the analysis.

Response: Thanks for the valuable advice! In the original manuscript, we had adjusted for a history of CVD or cancer and a history of hypertension in model 3, which included comorbidities such as heart attack, stroke, high blood pressure and so on. In the revised version, illness or injury in the past 2 years and longstanding illness or disability have been additionally adjusted for as suggested. The results did not materially change from the original ones, which will not alter our conclusions. The revised part was listed as follows:

[Methods (page 16; lines 304-312)] *“Our study finally included age (years between birth date and the start of accelerometry), sex, ethnicity, education, smoking status, alcohol intake (product of intake frequency and gram of each type of alcohol per one standard drink) (Biddinger, K. J. et al., JAMA Netw Open 2022), diet quality, sleep quality, self-rated health status, history of cancer, CVD, hypertension, long-standing illness, disability, or infirmity, and experience of serious illness, injury, bereavement, stress in last two years derived from the questionnaires, body mass index (BMI, kg/m², weight in kilograms divided by height in meters squared) and waist circumference (cm) measured during the initial visit, season and wear duration recorded by the accelerometers, and diabetes duration (years between the first*

occurrence of diabetes and the start of accelerometry).”

Comment 6: Line 99 “If current PA recommendations were followed, the risk of all-cause mortality might be reduced by..”, can this be shown visually? Again causal claims should be avoided.

Response: We appreciate the reviewer’s advice. As suggested, the causal claims have been removed and the association between categorical levels of PA and the risk of mortality has been visually presented as follows:

Figure 2. Association of accelerometer-derived physical activity with all-cause and cause-specific mortality among participants with type 2 diabetes.

HR indicates hazard ratio; CI, confidence interval; LPA, light-intensity physical activity; MPA, moderate-intensity physical activity; VPA, vigorous-intensity physical activity; MVPA, moderate-to-vigorous-intensity physical activity.

activity.

All adjusted for age (years), sex (male or female), ethnicity (white or others), education (college/university or others), season at the time of accelerometry recording (spring, summer, autumn, or winter), accelerometer wear duration (days), smoking status (never, former, or current), alcohol intake (g/day), diet score (0 to 7), sleep score (0 to 5), body mass index (kg/m²), waist circumference (cm), self-rated health (excellent, good, fair, or poor), long-standing illness, disability or infirmity (yes or no), illness, injury, bereavement, or stress in last 2 years (yes or no), history of cancer or cardiovascular disease (yes or no), history of hypertension (yes or no), and diabetes duration (years).

Comment 7:• In Statistical analysis section, Model 1 should be clearly written (if possible in forms of an equation), before extending to Model 2 and 3.

Response: As suggested, we have revised the manuscript to clearly illustrate the models used in our analysis.

[Methods (page 18; lines 338-348)] *“Three multivariable-adjusted models were constructed to account for potential confounding: model 1 was the crude model that adjusted for age (continuous, in years), sex (male or female), ethnicity (white or others), season (spring, summer, autumn, or winter) and duration of the accelerometry (continuous); model 2 was further adjusted for smoking status (never, former, or current), alcohol intake (continuous, in gram/day), diet scores (categories, 0 to 7), and sleep scores (continuous, 0 to 5); model 3 was the main model that additionally adjusted for BMI (continuous, in kg/m²), waist circumference (continuous, in cm), self-rated health (excellent, good, fair, or poor), long-standing illness, disability or infirmity (yes or no), illness, injury, bereavement, stress in last 2 years (yes or no), history of cancer or CVD (yes or no), history of hypertension (yes or no), and diabetes duration (continuous, in years).”*

Comment 8:• “Dose-response associations of PA with all-cause and cause-specific mortality were evaluated using restricted cubic splines fitted in the Cox proportional hazards models”, more details need to be presented on the estimation method and the software that was used to

fit the model, so that the analysis might be replicated.

Response: Thanks for the advice! More details of this process have been added to the revised manuscript as follows:

[Methods (page 17; lines 324-328)] *“Dose-response associations of PA with all-cause and cause-specific mortality were evaluated using restricted cubic splines fitted in the Cox proportional hazards models (“rms” package in R). The reference values were set at the 1st percentile, and knots at the 5th, 35th, 65th, and 95th percentiles of the PA distribution. Potential non-linearity was tested using a likelihood-ratio test comparing the Cox models with and without the cubic spline terms.”*

Comment 9:• In terms of the statistical modelling, Cancer, CVD mortality are competing risks. In a competing risks regression framework, Fine-Gray model might be an alternative to separate cause specific models, which could be explored as an additional analysis offering important insights.

Response: We thank the reviewer for this suggestion that may further strengthen our analysis. As suggested, we conducted an additional sensitivity analysis using the Fine-Gray competing risks regression model. In the analysis, we incorporated other-cause death as a competing risk when investigating the association between physical activity and cause-specific mortality. We found the results from the Fine-Gray models were consistent with our main findings, lending additional robustness to our conclusions (**Supplementary Table S21**). We revised the manuscript as follows:

[Methods (page 20; lines 379-381)] *“Tenth, we used the Fine and Gray subdistribution hazard model to examine the associations between PA and mortality, incorporating CVD death or cancer death as a competing risk.”*

[Results (page 8; line 137-144)] *“The results were largely consistent with the main analyses when ... when the competing risk regression model was used (**Supplementary Table S21**) ...”*

Table S21. Association of physical activity with cancer and cardiovascular disease mortality risk using Fine & Gray models for competing risk (n = 19,624) *

Outcomes	LPA (minutes/week)			
	<1750	1750-2099	2100-2449	≥2450
Cancer mortality				
HR (95% CI)	1 (ref)	0.82 (0.64,1.03)	0.67 (0.51,0.87)	0.75 (0.55,1.02)
Cardiovascular disease mortality				
HR (95% CI)	1 (ref)	0.81 (0.55,1.19)	0.90 (0.60,1.35)	0.78 (0.45,1.35)
Outcomes	MPA (minutes/week)			
	<150	150-299	300-449	≥450
Cancer mortality				
HR (95% CI)	1 (ref)	0.70 (0.52,0.95)	0.64 (0.47,0.88)	0.49 (0.35,0.69)
Cardiovascular disease mortality				
HR (95% CI)	1 (ref)	0.89 (0.58,1.38)	0.55 (0.33,0.91)	0.53 (0.31,0.90)
Outcomes	VPA (minutes/week)			
	0	1-14	15-29	≥30
Cancer mortality				
HR (95% CI)	1 (ref)	0.71 (0.56,0.91)	0.72 (0.55,0.93)	0.65 (0.47,0.91)
Cardiovascular disease mortality				
HR (95% CI)	1 (ref)	0.73 (0.50,1.07)	0.77 (0.52,1.16)	0.32 (0.16,0.62)
Outcomes	MVPA (minutes/week)			
	<275	275-449	450-624	≥625
Cancer mortality				
HR (95% CI)	1 (ref)	0.63 (0.50,0.81)	0.71 (0.54,0.93)	0.52 (0.37,0.72)
Cardiovascular disease mortality				

mortality				
HR (95% CI)	1 (ref)	0.58 (0.40,0.85)	0.58 (0.37,0.90)	0.31 (0.16,0.61)

HR indicates hazard ratio; CI, confidence interval; LPA, light-intensity physical activity; MPA, moderate-intensity physical activity; VPA, vigorous-intensity physical activity; MVPA, moderate-to-vigorous-intensity physical activity.

* Hazard ratios were calculated in Cox proportional hazards model after adjusting for age (years), sex (male or female), ethnicity (white or others), education (college/university or others), season at the time of accelerometry recording (spring, summer, autumn, or winter), accelerometer wear duration (days), smoking status (never, former, or current), alcohol intake (g/day), diet score (0 to 7), sleep score (0 to 5), body mass index (kg/m²), waist circumference (cm), self-rated health (excellent, good, fair, or poor), long-standing illness, disability or infirmity (yes or no), illness, injury, bereavement, or stress in last 2 years (yes or no), history of cancer or cardiovascular disease (yes or no), history of hypertension (yes or no), and diabetes duration (years).

Minor Concerns

Comment 10: • In terms of assessing possible selection bias from missing data, supplementary table S4 shows the characteristics of participants with type 2 diabetes included and excluded from the current study. The findings from this table should be mentioned in the main paper.

Response: Thank you for your useful suggestion. We have added findings from the supplementary table S4 in the main paper. Specifically, patients with type 2 diabetes included in the analyses were generally healthier than those who were excluded. Accordingly, we acknowledge in the Limitations section that this may have led to some degree of selection bias. The revised version was listed as follows:

[Results (page 5; lines 79-82)] *“Diabetic participants included in the analyses were generally healthier than those who were excluded, with a higher level of education, a smaller body mass index (BMI) and waist circumference, a lower prevalence of smoking, and better health status at baseline (Supplementary Table S1).”*

[Discussion (page 13; lines 249-256)] *“Sixth, though the sample size of the UK Biobank*

was as large as 0.5 million, the response rate was only 5.5% and accelerometer-measured PA was only available for a subset of participants. Diabetic adults in these analyses were generally healthier, more physically active, and had a lower mortality rate than the overall diabetic adults. The selection bias may limit the generalizability of the current findings. Nonetheless, recent evidence suggests that any potential bias resulting from poor representativeness would have a minimal impact on the association of PA with mortality in the UK Biobank (Stamatakis E, et al., *Epidemiology* 2021).”

Comment 11: P might be replaced by “p-value” throughout the paper. Importantly, the actual p-value needs to be reported in the main paper unless they are <0.001 (very small).

Response: Thanks for the comment! All “P” values throughout the paper have been replaced by “p-value” followed by actual values except that 36 actual p-values for interaction that were all greater than 0.05 in the stratified analyses were specified only in the supplementary material.

Comment 12: The proportional hazard assumption diagnostics needs to be presented in the supplementary material, at least for the main model.

Response: According to the reviewer’s suggestion, we have included the proportional hazards assumption diagnostics for our main models in Supplementary Table S27. All P-values were greater than 0.05, indicating that the proportional hazards assumption was not violated.

[**Methods (page 18; lines 348-350)**] “*The proportional hazard assumption for Cox models was checked with Schoenfeld residuals and no violation was observed (Supplementary Table S27).*”

Table S27. Proportional hazard test for the main model

Model 3 in the primary analysis		Proportional hazard test for global model
Exposures	Outcomes	P values
LPA (minutes/week)	All-cause mortality	0.2644

MPA (minutes/week)	All-cause mortality	0.1818
VPA (minutes/week)	All-cause mortality	0.1568
MVPA (minutes/week)	All-cause mortality	0.2997
LPA (minutes/week)	Cancer-cause mortality	0.0541
MPA (minutes/week)	Cancer-cause mortality	0.0908
VPA (minutes/week)	Cancer-cause mortality	0.2513
MVPA (minutes/week)	Cancer-cause mortality	0.0535
LPA (minutes/week)	CVD-cause mortality	0.3501
MPA (minutes/week)	CVD-cause mortality	0.4265
VPA (minutes/week)	CVD-cause mortality	0.4631
MVPA (minutes/week)	CVD-cause mortality	0.4749

Comment 13:• State (name) and refer to the testing method used for testing non linearity: “P for non-linearity <0.05”

Response: We thank the reviewer for this suggestion. Accordingly, we have stated the testing method for non-linearity in the revised version as follows:

[Methods (page 17; lines 327-328)] *“Potential non-linearity was tested using a likelihood-ratio test comparing the Cox models with and without the cubic spline terms.”*

Comment 14:• “Then, the duration of PA was categorized into five levels mainly following the cutoff points defined by the current PA recommendations.” Be explicit above the cut-off points and/or cite relevant literature.

Response: Since the number of cases in the fifth category was so small that may affect the statistical power, in the revised manuscript, the duration of PA was re-categorized into four levels as suggested by another reviewer. Specific cut-off points and the reasons for classification have been added to the text as follows:

[Methods (page 17; lines 328-335)] *“Then, the duration of PA was categorized into four levels: <1750, 1750-2099, 2100-2449, and ≥2450 minutes/week for LPA, <150, 150-299, 300-*

449, and ≥ 450 minutes/week for MPA, 0, 1-14, 15-29, and ≥ 30 minutes/week for VPA, and < 275 , 275-449, 450-624, and ≥ 625 minutes/week for MVPA after considering recommendations in the current PA guideline (≥ 150 minutes/week of MPA or ≥ 75 minutes/week of VPA), the inflection points of the dose-response relationship in this study (about 1750, 400, 15, and 375 minutes/week for LPA, MPA, VPA, and MVPA), and the sample size of each group.”

Comment 15:• Thorough proofreading is recommended.

Response: We thank the reviewer for this recommendation. To ensure a high quality of the final manuscript, we have thoroughly proofread the revised version, paying close attention to grammar, spelling, punctuation, and consistency of formatting.

REVIEWER COMMENTS

Reviewer #1 (Remarks to the Author):

I have no further comments and suggestions for the authors.

Reviewer #2 (Remarks to the Author):

The authors have addressed my comments sufficiently except for 2 points:

1) Given the lack of generalizability of people with T2D in this study of the UK biobank, would favor addingin the UK Biobank.. to the title of the paper. Based on the data presented, we cannot generalize to all people with T2D.

2) The second comment relates to:

"As high as 38.6% of death could be prevented if T2D patients performed 159 ≥625 minutes/week of MVPA"

This sentence still makes causal claims, or at least suggests that, and I felt that this study does not conclusively supports this claim.

Reviewer #3 (Remarks to the Author):

The authors do a good job addressing the concerns. I have no further comments.

Reviewer #4 (Remarks to the Author):

The authors have done a commendable job in revising the manuscript.

However, I believe, the results are not new and have been established in previous studies. I have a few comments and concerns in regards to a few of the responses provided by the authors. Please see the attached report.

Review

The authors have provided their answers to the queries raised in the previous round and done a commendable job in revising the manuscript. I have the following comments and concerns in regards to a few of the responses provided by the authors.

Major Concerns

- **Comment 1:** The article’s purpose has been well justified in the revised article. However the aim mentioned in this study has been explored before, in particular, the authors fail to mention (Bakrania et al. 2017; Yerramalla et al. 2020) and similar other studies.
- **Comment 2:** The authors mention all causal claims have been removed in the revised article. The sentences “As high as 38.6% of death could be prevented if T2D patients performed ≥ 625 minutes/week of MVPA” suggest a causal direction.
- **Comment 3:**

The lack of literature review in in terms of already existing work on objectively measured PA and all cause, CVD (Ledbetter et al. 2022) (> 3000 individuals) or cancer mortality (Leroux et al. 2021) (a UK biobank study with > 80000 individuals) still persists in the revised paper and is concerning. So saying “However, their sample sizes were less than 2000; accelerometers were worn on the hip ...” is factually incorrect.
- **Comment 4:** Please cite appropriate literature.
- **Comment 9:** The authors mention “Tenth, we used the Fine and Gray sub-distribution hazard model to examine the associations between PA and mortality, incorporating CVD death or cancer death as a competing risk.” Please clarify exactly how were the outcomes defined (and who were considered to be censored) in this model.
- **Comment 10:** The authors mention “Specifically, patients with type 2 diabetes included in the analyses were generally healthier than those who were excluded. Accordingly, we acknowledge in the Limitations section that this may have led to some degree of selection bias”, “Diabetic adults in these analyses were generally healthier, more physically active, and had a lower mortality rate than the overall diabetic adults. The selection bias may limit the generalizability of the current findings.” What could be possible ways to mitigate this selection bias (future work)?

- **Comment 12:** The authors mention that the proportional hazard assumption for Cox models was checked with Schoenfeld residuals and no violation was observed. Is this also true for other non PA covariates?
- **Comment 13:** I am wondering how the cut-points of different activity type was determined. This generally varies from device to device and population (different age groups). The authors mention “Minutes per week of LPA, MPA, and VPA were determined as the time spent in 30 to 125 milligravities (mg), >125 to 400 mg, and > 400 mg intensity activity, respectively [23]”. Is this standard practice for UK Biobank? Leroux et al. (2021) defines LPA and MVPA were determined as the time spent in 30 to 100 milligravities (mg) and >100 mg respectively. Some comment on the subjectivity introduced by the choice of these thresholds should be made.

References

- Kishan Bakrania, Charlotte L Edwardson, Kamlesh Khunti, Joseph Henson, Emmanuel Stamatakis, Mark Hamer, Melanie J Davies, and Thomas Yates. Associations of objectively measured moderate-to-vigorous-intensity physical activity and sedentary time with all-cause mortality in a population of adults at high risk of type 2 diabetes mellitus. *Preventive medicine reports*, 5:285–288, 2017.
- Mark K Ledbetter, Lucia Tabacu, Andrew Leroux, Ciprian M Crainiceanu, and Ekaterina Smirnova. Cardiovascular mortality risk prediction using objectively measured physical activity phenotypes in nhanes 2003–2006. *Preventive medicine*, 164:107303, 2022.
- Andrew Leroux, Shiyao Xu, Prosenjit Kundu, John Muschelli, Ekaterina Smirnova, Nilanjan Chatterjee, and Ciprian Crainiceanu. Quantifying the predictive performance of objectively measured physical activity on mortality in the uk biobank. *The Journals of Gerontology: Series A*, 76(8):1486–1494, 2021.
- Manasa S Yerramalla, Aurore Fayosse, Aline Dugravot, Adam G Tabak, Mika Kivimäki, Archana Singh-Manoux, and Séverine Sabia. Association of moderate and vigorous physical activity with incidence of type 2 diabetes and subsequent mortality: 27 year follow-up of the whitehall ii study. *Diabetologia*, 63:537–548, 2020.

****Responses to the Reviewer #1****

Comment 1: I have no further comments and suggestions for the authors.

Response: We appreciate your support and recognition of our work.

****Responses to the Reviewer #2****

Comment 1: The authors have addressed my comments sufficiently except for 2 points: Given the lack of generalizability of people with T2D in this study of the UK biobank, would favor addingin the UK Biobank.. to the title of the paper. Based on the data presented, we cannot generalize to all people with T2D.

Response: Thank you for your insightful comments. We have carefully considered your suggestions regarding the generalizability of our study population from the UK Biobank. In accordance with your advice, we have amended the title of our paper to more accurately reflect the scope of our study.

The revised title is now: *“Association of accelerometer-derived physical activity with all-cause and cause-specific mortality among individuals with type 2 diabetes in the UK Biobank”*

Comment 2: The second comment relates to: "As high as 38.6% of death could be prevented if T2D patients performed ≥ 625 minutes/week of MVPA". This sentence still makes causal claims, or at least suggests that, and I felt that this study does not conclusively supports this claim.

Response: Thank you for pointing out this issue. We have revised this sentence to avoid the causal claim as follows:

[Discussion (page 9; lines 160-161)] *“Up to 38.6% of deaths among T2D patients could potentially be associated with performing < 625 minutes/week of MVPA.”*

****Responses to the Reviewer #3****

Comment 1: The authors do a good job addressing the concerns. I have no further comments.

Response: We appreciate your positive comments.

****Responses to the Reviewer #4****

Comment 1: The authors have provided their answers to the queries raised in the previous round and done a commendable job in revising the manuscript. I have the following comments and concerns in regards to a few of the responses provided by the authors.

Major Concerns

The article's purpose has been well justified in the revised article. However the aim mentioned in this study has been explored before, in particular, the authors fail to mention (Bakrania et al. 2017; Yerramalla et al. 2020) and similar other studies.

Response: We are grateful for your recognition of the improvements made in our manuscript and value your ongoing feedback. According to your kind reminders, we have now included these key studies examining the associations between PA and mortality in T2D patients, despite the limitations of self-reported PA or small sample size. We believe this addition in the Introduction section could elucidate the objective of our study more clearly. The revised part is listed as follows:

[Introduction (page 3; lines 46-48)] *“Moreover, some prospective cohort studies reported that ≥ 150 minutes/week of moderate-to-vigorous-intensity PA (MVPA) or ≥ 75 minutes/week of vigorous-intensity PA (VPA) was associated with 14%–37% lower risks of all-cause mortality in diabetic adults (Yerramalla, M. S. et al., Diabetologia 2020; Han, H. et al., Diabetes Care 2022; Jung, I. et al., Diabetes Res Clin Pract 2023).”*

[Introduction (page 4; lines 59-63)] *“Nevertheless, accelerometry-based evidence in diabetic adults was scant. Two studies derived from the NHANES, a study from the Look AHEAD Trial and a study from the Walking Away from Type 2 Diabetes trial all demonstrated significant risk reductions in related to all-cause mortality with higher volumes of PA (Loprinzi, P. D., Physiol Behav 2016; Zhu, P. et al., Int J Behav Nutr Phys Act 2023; Look AHEAD Study Group, Diabetes Care 2022; Bakrania, K. et al., Preventive Medicine Reports 2017).”*

Comment 2: The authors mention all causal claims have been removed in the revised article. The sentences “As high as 38.6% of death could be prevented if T2D patients performed ≥ 625

minutes/week of MVPA” suggest a causal direction.

Response: Thank you for pointing out this issue. As suggested, we have revised this sentence to avoid the causal claim as follows:

[Discussion (page 9; lines 160-161)] *“Up to 38.6% of deaths among T2D patients could potentially be associated with performing <625 minutes/week of MVPA.”*

Comment 3: The lack of literature review in in terms of already existing work on objectively measured PA and all cause, CVD (Ledbetter et al. 2022) (> 3000 individuals) or cancer mortality (Leroux et al. 2021) (a UK biobank study with > 80000 individuals) still persists in the revised paper and is concerning. So saying “However, their sample sizes were less than 2000; accelerometers were worn on the hip ...” is factually incorrect.

Response: Thank you for pointing out these issues. According to your comments, we have thoroughly checked the Introduction section and included an expanded literature review in the revised manuscript, incorporating key studies such as Ledbetter et al. 2022 and Leroux et al. 2021. These additions provide a more comprehensive overview of the existing work related to objectively measured PA and its impact on all-cause, CVD, and cancer mortality. Furthermore, we have corrected the factual inaccuracies in the original manuscript to clarify the limitations of the previous accelerometry-based studies in diabetic adults more exactly. The revised manuscript was listed as follows:

[Introduction (page 4; lines 57-63)] *“In addition, existing studies have demonstrated that objectively measured PA could be a potential predictor for CVD mortality and cancer mortality in the general population (Leroux, A. et al., J Gerontol A Biol Sci Med Sci 2021; Ledbetter, M. K. et al., Prev Med 2022). Nevertheless, accelerometry-based evidence in diabetic adults was scant. Two studies derived from the NHANES, a study from the Look AHEAD Trial and a study from the Walking Away from Type 2 Diabetes trial all demonstrated significant risk reductions in related to all-cause mortality with higher volumes of PA (Loprinzi, P. D., Physiol Behav 2016; Zhu, P. et al., Int J Behav Nutr Phys Act 2023; Look AHEAD Study Group, Diabetes Care 2022; Bakrania, K. et al., Preventive Medicine Reports 2017). However,*

the sample sizes of these accelerometry-based research among T2D adults were less than 2000; accelerometers were worn on the hip or waist, which may omit activities that primarily involve upper body movement; and dose-response analyses were not performed.”

Comment 4: Please cite appropriate literature.

Response: Thank you for your valuable suggestion. As suggested, we have thoroughly reviewed our manuscript and have now incorporated additional citations that align closely with our study’s aims and findings.

Comment 5: The authors mention “Tenth, we used the Fine and Gray sub-distribution hazard model to examine the associations between PA and mortality, incorporating CVD death or cancer death as a competing risk.” Please clarify exactly how were the outcomes defined (and who were considered to be censored) in this model.

Response: Thank you for your insightful comments. In our analyses using the Fine and Gray competing risk regression models, we focused on CVD mortality and the cancer mortality as the outcomes of interest. We accounted for cancer deaths as a competing risk in examining the association between PA and CVD mortality. Likewise, CVD deaths were considered as a competing risk in the analysis of the relationship between PA and cancer mortality. To clarify this information exactly, we revised the manuscript as follows:

[Methods (page 20-21; lines 391-395)] *“Tenth, we employed the Fine and Gray sub-distribution hazard model to examine the associations between PA and CVD mortality, accounting for cancer deaths as a competing risk. Similarly, we considered CVD deaths as a competing risk when analyzing the relationship between PA and cancer mortality through competing risk analyses.”*

Comment 6: The authors mention “Specifically, patients with type 2 diabetes included in the analyses were generally healthier than those who were excluded. Accordingly, we acknowledge in the Limitations section that this may have led to some degree of selection bias”, “Diabetic

adults in these analyses were generally healthier, more physically active, and had a lower mortality rate than the overall diabetic adults. The selection bias may limit the generalizability of the current findings.” What could be possible ways to mitigate this selection bias (future work)?

Response: Thank you for highlighting the potential issue of selection bias in our study. To address this concern, future research ought to investigate the associations between PA and mortality within T2D patients across a broader range of national populations, ensuring a longer follow-up time and larger sample sizes of diabetic adults. Accordingly, we revised the manuscript as follows:

[Methods (page 14; lines 260-264)] *“The selection bias may limit the generalizability of the current findings. Future studies are recommended to verify these associations between PA and mortality in T2D patients across various national populations, with longer follow-up time and larger sample size of diabetic adults.”*

Comment 7: The authors mention that the proportional hazard assumption for Cox models was checked with Schoenfeld residuals and no violation was observed. Is this also true for other non PA covariates?

Response: Thank you for pointing out this issue. Following your suggestion, we re-checked our Cox proportional hazards models. In the reassessment, we identified all non-PA covariates meet the proportional hazard assumption except for age and history of CVD and cancer. To rectify this, we have introduced interaction terms between these covariates and time in our model. We are pleased to report that the outcomes of these Cox regression model with time-varying covariates remain consistent with our previous findings, affirming that these changes do not alter the conclusions of this study. In the revised version, we have clarified this information in the Methods section:

[Methods (page 19; lines 358-360)] *“In the primary analyses, age and history of cancer or CVD were treated as time-varying covariates by constructing interaction terms between covariates and time.”*

Table 2. Hazard ratios (95% CI) for all-cause mortality according to physical activity among individuals with type 2 diabetes

Exposures	No. of cases	Incidence rate per 1000 person-	Hazard ratio (95% CI) *		
			Model 1	Model 2	Model 3
LPA (minutes/week)					
<1750	336	9.58	1 (Ref)	1 (Ref)	1 (Ref)
1750-2099	229	5.62	0.66 (0.56,0.78)	0.68 (0.58,0.81)	0.77 (0.65,0.92)
2100-2449	149	4.27	0.53 (0.44,0.65)	0.55 (0.45,0.67)	0.65 (0.53,0.79)
≥2450	96	4.02	0.54 (0.43,0.68)	0.56 (0.44,0.71)	0.68 (0.54,0.87)
p-value for trend			<0.001	<0.001	<0.001
MPA (minutes/week)					
<150	182	23.56	1 (Ref)	1 (Ref)	1 (Ref)
150-299	255	9.40	0.47 (0.39,0.57)	0.50 (0.41,0.60)	0.57 (0.47,0.70)
300-449	190	5.20	0.31 (0.26,0.39)	0.34 (0.27,0.42)	0.42 (0.34,0.53)
≥450	183	2.90	0.22 (0.18,0.27)	0.24 (0.19,0.30)	0.32 (0.25,0.40)
p-value for trend			<0.001	<0.001	<0.001
VPA (minutes/week)					
0	322	13.36	1 (Ref)	1 (Ref)	1 (Ref)
1-14	229	5.90	0.52 (0.44,0.62)	0.54 (0.46,0.64)	0.61 (0.52,0.73)
15-29	172	4.61	0.45 (0.37,0.54)	0.47 (0.39,0.57)	0.57 (0.47,0.70)
≥30	87	2.53	0.30 (0.24,0.38)	0.33 (0.26,0.42)	0.42 (0.33,0.55)
p-value for trend			<0.001	<0.001	<0.001
MVPA (minutes/week)					
<275	394	14.27	1 (Ref)	1 (Ref)	1 (Ref)
275-449	202	5.33	0.46 (0.39,0.55)	0.49 (0.41,0.58)	0.55 (0.47,0.66)
450-624	134	3.92	0.40 (0.33,0.49)	0.42 (0.35,0.52)	0.51 (0.42,0.63)
≥625	80	2.29	0.29 (0.22,0.37)	0.31 (0.24,0.40)	0.39 (0.30,0.50)
p-value for trend			<0.001	<0.001	<0.001

HR indicates hazard ratio; CI, confidence interval; LPA, light-intensity physical activity; MPA, moderate-intensity physical activity; VPA, vigorous-intensity physical activity; MVPA,

moderate-to-vigorous-intensity physical activity.

* Hazard ratios (95% CI) were calculated in Cox proportional hazards model: model 1, adjusted for age (years), sex (male or female), ethnicity (white or others), education (college/university or others), season at the time of accelerometry recording (spring, summer, autumn, or winter), and accelerometer wear duration (days); model 2, further adjusted for smoking status (never, former, or current), alcohol intake (g/day), diet score (0 to 7), and sleep score (0 to 5) based on model 1; model 3, further adjusted for body mass index (kg/m²), waist circumference (cm), self-rated health (excellent, good, fair, or poor), long-standing illness, disability or infirmity (yes or no), illness, injury, bereavement, or stress in last 2 years (yes or no), history of cancer or cardiovascular disease (yes or no), history of hypertension (yes or no), and diabetes duration (years) based on model 2.

Table S6. Hazard ratios (95% CI) for cancer mortality according to physical activity among individuals with type 2 diabetes

Exposures	Cancer mortality No. of cases	Incidence rate per 1000 person-year	HR (95% CI) *		
			Model 1	Model 2	Model 3
LPA					
(minutes/week)					
<1750	168	4.79	1 (ref)	1 (ref)	1 (ref)
1750-2099	129	3.16	0.72 (0.57,0.91)	0.75 (0.59,0.94)	0.80 (0.64,1.02)
2100-2449	84	2.41	0.57 (0.44,0.75)	0.59 (0.45,0.77)	0.65 (0.50,0.86)
≥2450	60	2.51	0.62 (0.46,0.85)	0.65 (0.48,0.88)	0.73 (0.54,1.00)
p-value for trend			<0.001	<0.001	=0.009
MPA					
(minutes/week)					
<150	72	9.32	1 (ref)	1 (ref)	1 (ref)
150-299	128	4.72	0.59 (0.44,0.79)	0.62 (0.46,0.83)	0.67 (0.50,0.91)
300-449	120	3.28	0.48 (0.36,0.65)	0.52 (0.39,0.71)	0.59 (0.43,0.81)
≥450	121	1.91	0.35 (0.26,0.47)	0.38 (0.28,0.52)	0.44 (0.31,0.61)
p-value for trend			<0.001	<0.001	<0.001

VPA					
(minutes/week)					
0	149	6.18	1 (ref)	1 (ref)	1 (ref)
1-14	125	3.22	0.61 (0.48,0.78)	0.63 (0.50,0.81)	0.68 (0.53,0.87)
15-29	102	2.73	0.57 (0.44,0.74)	0.60 (0.47,0.78)	0.67 (0.51,0.88)
≥30	65	1.89	0.48 (0.36,0.65)	0.52 (0.38,0.71)	0.58 (0.42,0.81)
p-value for trend			<0.001	<0.001	=0.038
MVPA					
(minutes/week)					
<275	183	6.63	1 (ref)	1 (ref)	1 (ref)
275-449	111	2.93	0.53 (0.42,0.68)	0.56 (0.44,0.71)	0.60 (0.47,0.77)
450-624	93	2.72	0.58 (0.45,0.74)	0.61 (0.47,0.79)	0.66 (0.51,0.87)
≥625	54	1.54	0.39 (0.29,0.54)	0.42 (0.31,0.58)	0.46 (0.33,0.64)
p-value for trend			<0.001	<0.001	<0.001

HR indicates hazard ratio; CI, confidence interval; LPA, light-intensity physical activity; MPA, moderate-intensity physical activity; VPA, vigorous-intensity physical activity; MVPA, moderate-to-vigorous-intensity physical activity.

* Hazard ratios (95% CI) were calculated in Cox proportional hazards model: model 1, adjusted for age (years), sex (male or female), ethnicity (white or others), education (college/university or others), season at the time of accelerometry recording (spring, summer, autumn, or winter), and accelerometer wear duration (days); model 2, further adjusted for smoking status (never, former, or current), alcohol intake (g/day), diet score (0 to 7), and sleep score (0 to 5) based on model 1; model 3, further adjusted for body mass index (kg/m²), waist circumference (cm), self-rated health (excellent, good, fair, or poor), long-standing illness, disability or infirmity (yes or no), illness, injury, bereavement, or stress in last 2 years (yes or no), history of cancer or cardiovascular disease (yes or no), history of hypertension (yes or no), and diabetes duration (years) based on model 2.

Table S7. Hazard ratios (95% CI) for cardiovascular-disease mortality according to physical activity among individuals with type 2 diabetes

Exposures	Cardiovascular disease mortality No. of cases	Incidence rate per 1000 person-year	HR (95% CI) *		
			Model 1	Model 2	Model 3
LPA (minutes/week)					
<1750	73	2.08	1 (ref)	1 (ref)	1 (ref)
1750-2099	45	1.10	0.64 (0.44,0.92)	0.66 (0.46,0.96)	0.79 (0.54,1.15)
2100-2449	37	1.06	0.67 (0.45,1.00)	0.69 (0.46,1.03)	0.86 (0.57,1.30)
≥2450	17	0.71	0.52 (0.30,0.89)	0.54 (0.32,0.93)	0.74 (0.43,1.28)
			p-value for trend		
			=0.007	0.012	=0.327
MPA (minutes/week)					
<150	36	4.66	1 (ref)	1 (ref)	1 (ref)
150-299	64	2.36	0.61 (0.41,0.93)	0.66 (0.44,1.00)	0.84 (0.55,1.28)
300-449	35	0.96	0.31 (0.19,0.49)	0.34 (0.21,0.55)	0.50 (0.30,0.82)
≥450	37	0.59	0.25 (0.15,0.40)	0.28 (0.17,0.45)	0.45 (0.28,0.77)
			p-value for trend		
			<0.001	<0.001	=0.004
VPA (minutes/week)					
0	68	2.82	1 (ref)	1 (ref)	1 (ref)
1-14	51	1.31	0.54 (0.38,0.78)	0.57 (0.40,0.83)	0.69 (0.47,1.00)
15-29	42	1.12	0.49 (0.33,0.73)	0.53 (0.35,0.78)	0.71 (0.47,1.07)
≥30	11	0.32	0.17 (0.09,0.32)	0.19 (0.10,0.36)	0.28 (0.14,0.55)
			p-value for trend		
			<0.001	<0.001	=0.001
MVPA (minutes/week)					
<275	91	3.30	1 (ref)	1 (ref)	1 (ref)
275-449	42	1.11	0.43 (0.29,0.62)	0.45 (0.31,0.65)	0.55 (0.38,0.80)
450-624	28	0.82	0.37 (0.24,0.58)	0.40 (0.26,0.62)	0.54 (0.34,0.85)
≥625	11	0.31	0.18 (0.09,0.34)	0.19 (0.10,0.36)	0.28 (0.14,0.54)
			p-value for trend		
			<0.001	<0.001	<0.001

HR indicates hazard ratio; CI, confidence interval; LPA, light-intensity physical activity; MPA, moderate-

intensity physical activity; VPA, vigorous-intensity physical activity; MVPA, moderate-to-vigorous-intensity physical activity.

* Hazard ratios (95% CI) were calculated in Cox proportional hazards model: model 1, adjusted for age (years), sex (male or female), ethnicity (white or others), education (college/university or others), season at the time of accelerometry recording (spring, summer, autumn, or winter), and accelerometer wear duration (days); model 2, further adjusted for smoking status (never, former, or current), alcohol intake (g/day), diet score (0 to 7), and sleep score (0 to 5) based on model 1; model 3, further adjusted for body mass index (kg/m²), waist circumference (cm), self-rated health (excellent, good, fair, or poor), long-standing illness, disability or infirmity (yes or no), illness, injury, bereavement, or stress in last 2 years (yes or no), history of cancer or cardiovascular disease (yes or no), history of hypertension (yes or no), and diabetes duration (years) based on model 2.

Comment 8: I am wondering how the cut-points of different activity type was determined. This generally varies from device to device and population (different age groups). The authors mention “Minutes per week of LPA, MPA, and VPA were determined as the time spent in 30 to 125 milligravities (mg), >125 to 400 mg, and > 400 mg intensity activity, respectively [23]”. Is this standard practice for UK Biobank? Leroux et al. (2021) defines LPA and MVPA were determined as the time spent in 30 to 100 milligravities (mg) and >100 mg respectively. Some comment on the subjectivity introduced by the choice of these thresholds should be made.

References

Kishan Bakrania, Charlotte L Edwardson, Kamlesh Khunti, Joseph Henson, Emmanuel Stamatakis, Mark Hamer, Melanie J Davies, and Thomas Yates. Associations of objectively measured moderate-to-vigorous-intensity physical activity and sedentary time with all-cause mortality in a population of adults at high risk of type 2 diabetes mellitus. *Preventive medicine reports*, 5:285–288, 2017.

Mark K Ledbetter, Lucia Tabacu, Andrew Leroux, Ciprian M Crainiceanu, and Ekaterina Smirnova. Cardiovascular mortality risk prediction using objectively measured physical activity phenotypes in nhanes 2003–2006. *Preventive medicine*, 164:107303, 2022.

Andrew Leroux, Shiyao Xu, Prosenjit Kundu, John Muschelli, Ekaterina Smirnova, Nilanjan Chatterjee, and Ciprian Crainiceanu. Quantifying the predictive performance of objectively measured physical activity on mortality in the uk biobank. *The Journals of Gerontology: Series A*, 76(8):1486–1494, 2021.

Manasa S Yerramalla, Aurore Fayosse, Aline Dugravot, Adam G Tabak, Mika Kivimäki, Archana Singh-Manoux, and S'everine Sabia. Association of moderate and vigorous physical activity with incidence of type 2 diabetes and subsequent mortality: 27 year follow-up of the whitehall ii study. *Diabetologia*, 63:537–548, 2020.

Response: We thank you for careful review and pointing out this issue. We completely agree with that these thresholds might vary from the type of device used and the demographic characteristics of the population, such as different age groups.

In our research, we adopted cut-points of 30, 125, and 400 milligravities (mg) for LPA, MPA, and VPA, respectively, which was in line with the methodology used in certain segments of the previously published UK Biobank studies (Frederick, K. et al., *Circulation* 2022; Fanny Petermann-Rocha. et al., *BMC Medicine* 2021; Boyang Xiang. et al., *Hypertension* 2023). However, we recognize that other studies leveraging accelerometry data from the UK Biobank may employ different thresholds for activity classification. For instance, Leroux et al. (2021) used cut-points of 30 to 100 mg and > 100 mg to define LPA and MVPA. In addition, Walmsley et al. (2021) developed a machine-learning-based method to classify movement behaviours including LPA, MVPA, sedentary behaviours, and sleep in wrist-worn accelerometer data.

Therefore, we acknowledge that the choice of these cut-points introduces a degree of subjectivity and may impact the comparability of findings across different studies. To address this concern, we have added a discussion in the limitation part as follows:

[Discussion (page 13; lines 246-251)] “Fourth, the cut-points for various PA intensities in this study are consistent with most studies involving accelerometry data from the UK Biobank. However, it is important to note that these thresholds, which can be subject to

variability due to device and demographic differences, may introduce a degree of subjectivity. Future research should, therefore, utilize more objective methods, such as data-driven approaches based on machine learning, to differentiate between PA intensities.”

REVIEWERS' COMMENTS

Reviewer #4 (Remarks to the Author):

The authors have done a commendable job in addressing the concerns in the previous round. I have no further comments for the authors.